



**Influence of climate variability, fire and phosphorus limitation on the**
**vegetation structure and dynamics in the Amazon-Cerrado border**
Emily Ane Dionizio da Silva[1], Marcos Heil Costa[1], Andrea Almeida Castanho[2], Gabrielle Ferreira Pires[1],
Beatriz Schwantes Marimon[3], Ben Hur Marimon-Junior[3], Eddie Lenza[3], Fernando Martins Pimenta[1],
Xiaojuan Yang[4], Atul K. Jain[5]
[1] Department of Agricultural Engineering, Federal University of Viçosa (UFV), Viçosa, MG, Brazil
[2] The Woods Hole Research Center, 149 Woods Hole Rd., Falmouth, MA USA
[3] Federal University of Mato Grosso, Nova Xavantina Campus, Nova Xavantina, MT, Brazil
[4] Oak Ridge National Laboratoty, Oak Ridge, TN, USA
[5] Department of Atmospheric Sciences, University of Illinois at Urbana-Champaign
Correspondence to: Emily Ane D. da Silva (emilyy.ane@gmail.com)



**Abstract**
Climate, fire and soil nutritional limitation are important elements that affect the vegetation
dynamics in areas of forest-savanna transition. In this paper, we use the dynamic vegetation model
INLAND to evaluate the influence of inter-annual climate variability, fire and phosphorus (P) limitation
on the Amazon-Cerrado transitional vegetation structure and dynamics. We assess how each
environmental factor affects the net primary production, leaf area index and aboveground biomass (AGB),
and compare the AGB simulations of observed AGB map. We used two climate datasets − the 1960-1990
average seasonal climate and the 1948 to 2008 inter-annual climate variability, two regional datasets of
total soil P content in soil, based on regional (field measurements) and global data and the INLAND fire
module. Our results show that inter-annual climate variability, P limitation and fire occurrence gradually
improve  simulated  vegetation  types  and  these  effects  are  not  homogeneous  along  the
latitudinal/longitudinal gradient showing a synergistic effect among them. In terms of magnitude, the
effect of fire is stronger, and is the main driver of vegetation changes along the transition. The nutritional
limitation, in turn, is stronger than the effect of inter-annual climate variability acting on the transitional
ecosystems dynamics. Overall, INLAND typically simulates more than 80% of the AGB variability in
the transition zone. However, the AGB in many places is clearly not well simulated, indicating that
important soil and physiological factors in the Amazon-Cerrado border, such as lithology and water table
depth, carbon allocation strategies and mortality rates, still need to be included in the model.





## 1 Introduction

The Amazon and Cerrado are the two largest and most important phytogeographical domains in
South America. The Amazon forest has been globally recognized and distinguished not only for its
exuberance in diversity and species richness, but also for playing an important role in the global climate
by regulating water (Bonan, 2008; Pires and Costa, 2013) and heat fluxes (Shukla et al., 1990; Rocha et
al., 2004; Roy et al., 2002). The Cerrado is recognized worldwide for being the richest savanna in the
world (Myers et al., 2000; Klink and Machado, 2005). It is characterized by different physiognomies,
ranging from sparse physiognomies to dense woodland formations, and the latter are commonly mixed
with Amazon rainforest forming transitional areas. The Amazon-Cerrado transition extends for 6270 km
from northeast to southwest in Brazil, and the ecotonal vegetation around this transition is a mix of the
characteristics of the tropical forest and the savanna (Torello-Raventos et al., 2013).
Gradients of seasonal rainfall and water deficit, fire occurrence, herbivory and low fertility of the
soil have been reported as the main factors that characterize the transition between forest and savanna
globally (Lehmann et al., 2011; Hoffman et al., 2012; Murphy and Bowman, 2012). However, few studies
have evaluated the individual and combined effects of these factors on Brazilian ecosystems ecotones
(Marimon-Junior and Haridasan, 2005; Elias et al., 2013; Vourtilis et al., 2013).
It is challenging to assess the degree of interaction among these various environmental factors in
the transitional region and to infer how each one influences the distribution of the regional vegetation. In
this case, Dynamic Global Vegetation Models (DGVMs) can be powerful tools to isolate the influences
of climate, fire and nutrients, therefore helping to understand their large-scale effects on vegetation
(House et al., 2003; Favier et al., 2004; Hirota et al., 2010; Hoffman et al., 2012).



Previous modelling studies using DGVMs that investigate climate effects in the Amazon indicate

that the rainforest could experience changes in rainfall patterns which would either transform the forest
into an ecosystem with more sparse vegetation – similar to a savanna, what has been called as the
"savannization of the Amazon" (Shukla et al., 1990; Cox et al., 2000; Oyama and Nobre, 2003; Betts et
al., 2004; Cox et al., 2004; Salazar et al., 2007) - or to a seasonal forest (Malhi et al., 2009; Pereira et al.,
2012; Pires and Costa, 2013). These studies had great importance to the improvement of terrestrial
biosphere modeling, but they neglect two important processes in tropical ecosystem dynamics: fire
occurrence and nutrient limitation, particularly the P limitation.

In tropical ecosystems, fire plays an important ecological role and influences the productivity, the

biogeochemical cycles and the dynamics in the transitional biomes, not only by changing the phenology
and physiology of plants, but also by modifying the competition among trees and lower canopy plants
such as grasses, shrubs and lianas. Fire occurrence, depending on its frequency and intensity, may increase
the mortality of trees and transform an undisturbed forest into a disturbed and flammable one (House et
al., 2003; Hirota et al., 2010; Hoffmann et al., 2012). Fires also affect the dynamics of nutrients in the
savanna ecosystem, changing mainly the N:P relationship and P availability in the soil (Nardoto et al.

2006).

Studies suggest that P is the main limiting nutrient within tropical forests (Malhi et al., 2009;

Mercado et al., 2011; Quesada et al., 2012) unlike the temperate forests. Phosphorus (P) is a nutrient that
is easily adsorvided by soil minerals due to the large amount of iron and aluminum oxides in the Amazon
and Cerrado acidic and strongly weathered soils (Dajoz, 2005; Goedert, 1986). In the tropics, the warm
and wet climate favors the high biological activity in the soil and the litter decomposition, not limiting





the nitrogen for plant fixation. In Cerrado, higher soil fertility is related to regions with greater woody
plants abundance and less grass cover, similarly to the features found in the Amazon rainforest (Moreno
et al., 2008; Vourtilis et al., 2013; Veenendaal et al., 2015). However, the phosphorus limitation is often
neglected by DGVMs. which usually assume unlimited P availability and consider nitrogen as the main
limiting nutrient. However, N is not a limiting nutrient for trees in the tropics (Davidson et al. 2004),
while P availability affects the trees dynamics.
In principle, in transitional forests, where the climate is intermediate between wet and seasonally
dry, the heterogeneous structure and phenology make it difficult to represent these forests in models. The
Amazon-Cerrado border is the result of the expansion and contraction of the Cerrado into the forest (see
Marimon et al., 2006; Morandi et al., 2016), especially in the Mato Grosso state, where extreme events,
such as intense droughts, influence the vegetation dynamics (Marimon et al., 2014) and the nutrient
(Oliveira et al., *in press*) and carbon cycling (Valadão et al., 2016).
Currently, no model has demonstrated to be able to accurately simulate the vegetation transition
between Amazon and Cerrado. A better understanding of the main drivers that determine the distribution
of different vegetation physiognomies in the region is crucial for more reliable simulations of the
transitional tropical ecosystems in future climate scenarios.
In this paper we use the dynamic vegetation model INLAND (Integrated Model of Land Surface
Processes) to evaluate the influence of inter-annual climate variability, fire occurrence and P limitation
in the Amazon-Cerrado transitional vegetation dynamics and structure. We assess how each element
affects the net primary production (NPP), leaf area index (LAI) and aboveground biomass (AGB) and
compare the model simulated AGB to observed AGB data. The results presented here are important to



build models that accurately represent the transition vegetation, and show the need to include the spatial
variability of eco-physiological parameters in these areas.
**2      Materials and methods**
**2.1     Study Area**
The present study focuses on the Amazon-Cerrado transition (Figure 1). We use the official
delimitation of the Brazilian biomes proposed by IBGE (2004), and define five transects along the
transition border. Transects 1 to 4 are established considering approximately 330 km into the Amazon
and 330 km into the Cerrado domain, while Transect 5 is 880 km long on the southern Amazon-Cerrado
border. The transects are located as follows: Transect 1 (T1, 43°- 49°W; 5°- 7°S), Transect 2 (T2, 46°-
51° W; 7°-9S), Transect 3 (T3, 48°-54° W; 9°-11° S), Transect 4 (T4, 49° - 55° W; 11°-13° S), and
Transect 5 (T5, 53° - 61° W; 13°-15° S) (Figure 1).
**2.2     Description of the INLAND Surface Model**
The Integrated Model of Land Surface Processes (INLAND) is the land-surface component of the
Brazilian Earth System Model (BESM). INLAND is based on the IBIS model (Integrated Biosphere
Simulator, Foley et al., 1996; Kucharik et al., 2000), which considers changes in the composition and
structure of vegetation in response to the environment and incorporates important aspects of biosphere-
atmosphere interactions. The model simulates the exchanges of energy, water, carbon and momentum
between soil-vegetation-atmosphere. These processes are organized in a hierarchical framework and
operate at different time steps, ranging from 60 minutes to 1 year, coupling ecological, biophysical and
physiological processes. The vegetation structure is represented by two layers: upper (arboreal PFTs) and



lower (no arboreal PFTs, shrubs and grasses) canopies, and the composition is represented by 12 plant
functional types (PFTs) (e.g., tropical broadleaf evergreen trees or C4 grasses, among several others).The
photosynthesis and respiration processes are simulated in a mechanistic manner using the Ball-Berry-
Farquhar model (details in Foley et al., 1996). The vegetation phenology module simulates the processes
such as budding and senescence based on empirically-based temperature thresholds for each PFT. The
dynamic vegetation module computes the following variables yearly for each PFT: gross and net primary
productivity (GPP and NPP), changes in AGB pools, simple mortality disturbance processes and resultant
LAI, thus allowing vegetation type and cover to change with time. The partitioning of the NPP for each
PFT resolves carbon in three AGB pools: leaves, stems and fine roots. The LAI of each PFT is obtained
by simply dividing leaf carbon by specific leaf area, which in INLAND is considered fixed (one value)
for each PFT.

INLAND has eight soil layers to simulate the diurnal and seasonal variations of heat and moisture.

Each layer is described in terms of soil temperature, volumetric water content and ice content (Foley et
al., 1996; Thompson and Pollard, 1995). Furthermore, all of these processes are influenced by soil texture
and amount of organic matter within the soil profile.

Considering these aspects of vegetation dynamics and soil physical properties the model can

simulate plant competition for light and water between trees, shrubs and grasses through shading and
differences in water uptake (Foley et al., 1996). These PFTs can coexist within a grid cell and their annual
LAI values indicate the dominant vegetation type within a grid cell. For example, the dominant vegetation
type is a Tropical Evergreen Forest if the PFT tropical broadleaf evergreen tree has an annual mean upper
canopy LAI ($LAI_{upper}$) above 2.5 $m^2$ $m^{-2}$. On the other hand, the dominant vegetation type is a Tropical



Deciduous Forest if the tropical broadleaf drought-deciduous tree has an annual mean $LAI_{upper}$ above
2.5 $m^2$ $m^{-2}$. Where total tree LAI ($LAI_{upper}$) is between 0.8 and 2.5 $m^2$ $m^{-2}$, dominant vegetation type is
savanna, and $LAI_{upper}$ values smaller than 0.8 $m^2$ $m^{-2}$ characterize a grassland vegetation type.

We assume that the vegetation types Tropical Evergreen Forest and the Tropical Deciduous Forest

in INLAND represent the Amazon rainforest, while Savanna and Grasslands represent the Cerrado.
Savanna would be equivalent to the Cerrado physiognomies *Cerradão* and *Cerrado sensu strictu*, while
Grasslands would be equivalent to the physiognomies *Campo sujo* and *Campo Limpo* (*sensu* Ribeiro and
Walter, 2008).

The soil chemical properties are represented by the carbon, nitrogen and phosphorus. The carbon

cycle is simulated through vegetation, litter and soil organic matter, where the biogeochemical module is
similar to the CENTURY model (Parton et al., 1993; Verberne et al., 1990). The amount of C existing in
the first meter of soil is divided into different compartments characterized by their residence time, which
can vary in an interval of hours for microbial AGB and organic matter to several years for lignin. The
model considers only the soil N transformations and carbon decomposition, but the N cycle is not fully
simulated and N does not influence the vegetation productivity, i.e., there is a fixed C:N ratio. P is used
only to limit the gross primary productivity. The total P available in the soil ($P_{total}$) is used to estimate the
maximum capacity of carboxylation by the Rubisco enzyme ($V_{max}$) through a linear relationship.

$$V_{max} = 0.1013\,P_{total} + 30.037 \qquad\qquad\qquad (1)$$

where $V_{max}$ and $P_{total}$ are given in $\mu molCO_2$ $m^{-2}$ $s^{-1}$ and mg $kg^{-1}$, respectively. This equation has been
developed by Castanho et al. (2013) based on data for tropical evergreen and deciduous trees, and is
applied only to these two PFTs in the model.



INLAND also contains a fire module, from the Canadian Terrestrial Ecosystem Model CTEM
(Arora and Boer, 2005). In this module, three aspects of the fire triangle are considered – the availability
of fuel to burn, the flammability of vegetation depending on environmental conditions, and the presence
of an ignition source. The natural ignition probability is summed to arbitrary anthropogenic fire
probability, and the burned area is modeled as an ellipse of dimensions determined by wind and fuel
conditions (Arora and Boer, 2005).
**2.3    Observed data**
**2.3.1    Phosphorus databases**
We used two P databases to estimate $V_{max}$ (Equation 1): one regional (referred to as PR) and one
global database referred to as PG). In addition, a control P map (PC) represents the unlimited nutrient
availability case, equivalent to a $V_{max}$ of 65 $\mu molCO_2$ m$^{-2}$ s$^{-1}$, or 350 mg P kg$^{-1}$ soil, according to Equation

1.

The PR database was developed from total P in the soil for the Amazon basin published by
Quesada et al. (2011) plus 54 additional available P samples (P extracted via Mehlich-1 extractor,
$P_{mehlich-1}$) (Figure 2a). We used the $P_{-mehlich-1}$ and clay contents measured in a forest-savanna transition
region in Brazil (Mato Grosso state) to estimate $P_{total}$ and expand the coverage area of the P data (Section
S1). These 54 samples were gridded to a 1° × 1° grid to be compatible with the spatial resolution used by
INLAND, resulting in 12 additional pixels with observed total P content (Figure 2a). For pixels without
observed $P_{total}$, the $P_{total}$ was assumed to be 350 mg P kg$^{-1}$ soil, similarly to the PC conditions.



A global dataset of $P_{total}$ (Figure 2b) was also used to estimate $V_{max}$. This global data set is part of
a database containing six global maps of the different forms of P in the soil (Yang et al., 2013). The $P_{total}$
was estimated from lithologic maps, distribution of soil development stages, fraction of the remaining
source material for different stages of weathering using chronosequence studies (29 studies), and P
distribution in different forms for each soil type based on the analysis of Hedley fractionation (Yang and
Post, 2011), which are part of a worldwide collection of soil profile data. The uncertainties and limitations
associated with this database are restricted to the Hedley fractionation data used, which are 17% for low
weathered soils, 65% for intermediate soils and 68% for highly weathered soils (Yang et al., 2013).
**2.3.2    Above-Ground AGB (AGB) database**
The AGB database used was created by Nogueira et al. (2015) and considered undisturbed (pre-
deforestation) vegetation existing in the Brazilian Amazonia. This database was compiled from a
vegetation map at a scale of 1:250000 (IBGE, 1992) and AGB averages from 41 published studies that
had conducted direct sampling in either forest (2317 plots) or non-forest or contact zones (1830 plots).
We bi-linearly interpolated the AGB (dry weight) for each transect considering $1° \times 1°$ to ensure
compatibility of the observed and simulated data.
Five longitudinal transects (Figure 1) were used separately to characterize AGB in the Amazon-
Cerrado border (Figures 3a and 3b). In T1, T2, T3 and T4, the higher AGB values in the west and lower
values in the east are consistent with the transition from a dense and woody vegetation (the Amazon
forest) towards a sparse vegetation with lower AGB (the Cerrado). However, T1 shows a more gradual
reduction of AGB along the west to east gradient, while in T2, T3 and T4 where the transition is more





abrupt. In T5 no west-east gradient is present with high AGB heterogeneity and predominant low AGB
across the transect (Figure 3b).

**2.4    Simulations**

The model was forced with the prescribed climate data based on the Climate Research Unit (CRU)

database (Harris et al., 2014). Two climate boundary conditions were used: the first is referred to as the
monthly climatological average (CA) that represents the average climate for the period 1961-1990. The
second climate boundary condition is the historical dataset, for the continuous period between 1948 and
2008 (CV). For both boundary conditions, the variables used are rainfall, solar radiation, wind velocity
and maximum and minimum temperatures. The CRU database has been widely used by the scientific
community in case studies, because these data preserve the spatial mean of the rainfall data, although,
they do not provide adequate representation of their variance precipitation (Beguería et al., 2016). The
dataset has a 1-degree spatial resolution and a monthly time resolution.

Soil texture data is based on the IGBP-DIS global soil (Global Soil Data Task 2000) (Hansen and

Reed, 2000). The model simulations were run for the time period 1582-2008, a total of 427 years. In the
CV group of runs, the model was spin-up by cycling the 1948-2008 climate data (61-year) seven times,
totaling 427 years. In the CA group of runs, the annual mean climate data was cycled 427 times. In both
cases, $CO_2$ varied from 278 to 380 ppmv, according to observations in the period, updated annually. In
both cases, only the model results of the last 10 years were used to analyze the results.

The experiment design is a factorial combination of the climate scenarios (CA, monthly

climatological average, 1961-1990; CV, monthly climate time series, 1948-2008), the nutrient limitation



on $V_{max}$ (PC, no P limitation ($V_{max}$ = 65 $\mu molCO_2\, m^{-2}\, s^{-1}$); PR, regional P limitation; PG, global P
limitation) and the occurrence of fire (F) or not (Table 1). The 12 combinations in Table 1 allow the
evaluation of individual and combined effects of climate, soil chemistry, and the incidence of fire on the
variables: Net Primary Production (NPP), tree AGB, and LAI of the upper and lower canopies ($LAI_{upper}$,
$LAI_{lower}$).

We consider that the subtraction between the simulations (CV+PC) – (CA+PC) = (CV–CA)$|_{PC}$

represents the isolated effect of inter-annual climate variability without P limitations. The same logic is
applied to isolate other factors such as fire and P in different climate scenarios. For example, the fire
effect under average climate without P limitation case is calculated by the difference between CA+PC+F
and CA+PC, so that (CA+PC+F)–(CA+PC) = F$|_{CA,PC}$. Similarly, the isolated effect of fire under a climate
with inter-annual variability scenario without influence of P limitation is calculated by the difference
between CV+PC+F and CV+PC, so that (CV+PC+F)–(CV+PC) = F$|_{CV, PC}$. The different combinations of
climate scenarios with and without fire effects and with and without P limitations are described in Table

2.

**2.5   Statistical analysis and determination of the best model configuration**

The statistical analysis is divided in four parts. First, we present maps of the isolated effects for

all simulated area calculated as the average of last ten years of simulated spatial patterns. The statistical
significance of the isolated effects on NPP, LAI and AGB are determined using the t-test with $p < 0.05$.
The results are tested in each pixel, for all the simulated domain (n = 10).

Second, we present an analysis of variance using the one-way ANOVA and the Tukey-Kramer

test in the transition zone. We consider all 31 pixels which fall in transects T1 to T5 ($n_{pixels}$). The results



presented are based on the set of last 10 years of simulation (1999-2008, $n_{years}$) for the 12 combinations
($n_{simulation}$) in Table 1. Moreover, we grouped treatments according to climate regardless of P limitation,
presence or absence of fire, where all sets with CV vs CA are tested (Group 1, n=1860, ($n_{pixel}$ x $n_{year}$ x
($n_{simulation}$/2)). Similarly, in Group 2 we tested if PC, PR or PG were significantly different from each other
regardless the F or climate used (Group 2, n=1240, ($n_{pixel}$ x $n_{year}$ x ($n_{simulation}$/3)). In Group 3 we tested if
fire introduced a significant effect regardless of climate and P limitation (Group 3, n=1860, ($n_{pixel}$ x $n_{year}$
x ($n_{simulation}$/2)). Finally, all treatments were tested to each simulation assessing their individual effects on
NPP, LAI and AGB ($n_{pixel}$ x $n_{year}$ = 310).
Third, a correlation coefficient between the simulated and observed values for AGB was
calculated for each transect. The simulated variables are averaged for the last 10 years of simulations
(1999 - 2008) and compared to AGB from Nogueira et al. (2015) within a grid cell.
Finally, we evaluate INLAND's ability to assign the dominant vegetation type by analyzing 10
years of probability of occurrence. If the dominant vegetation type (evergreen tropical forest, or deciduous
forest for the Amazon rainforest, and savanna or grasslands for Cerrado) in a pixel is the same in more
than 90% of the simulated years (9 out of 10), then the simulated vegetation type is defined as "very
robust" for that pixel; if it occurs in 70 - 90% of the simulated years, the simulated result is considered to
be "robust". If the dominant vegetation occurred in less than 70% of simulated years, the pixel is
considered "transitional" vegetation.



## 3    Results

### 3.1    Influence of climate, fire and phosphorus in the Amazon-Cerrado transition region

#### 3.1.1    Spatial patterns

Overall, the inclusion of inter-annual climate variability (CV) resulted in a decrease in the simulated average tree biomass (TB) by 3.8% in Amazonia, and by 8.7% in Cerrado in comparison to average climate (CA) (Figure 4a). The spatial differences between CV and CA for TB simulations are statistically significant and range from -3 kg-C m$^{-2}$ to +2 kg-C m$^{-2}$. The state of Pará, with higher influence of the El Niño phenomenon, experienced the highest decrease in TB in the CV simulation. In the state of Roraima, on the other hand, there was an increase of about 2 kg-C m$^{-2}$ in TB when CV was considered. Bolivia and southwest of Mato Grosso state also presented, in some grids points, a significant increase in AGB higher than 2 kg-C m$^{-2}$.

On average, P acts as a limiting factor in the simulated TB, decreasing by 13% in regional P (PR) simulation and 15% in global P (PG) simulation. In PR, TB decreased mainly in the southeastern Amazonia (between Pará and northeastern Mato Grosso states) and northwestern Amazonas state (Figure 4b). In PG, the largest TB decline occurred in central Amazonia, northeastern Pará and northeastern Mato Grosso (Figure 4c). In Cerrado, on the other hand, TB declined by 2% for PR and 9% for PG with respect to the control simulation. In PR, the few pixels in the Cerrado that have P limitation showed a significant decrease in TB (Figure 4b), while in PG the TB reduction was statistically significant for most of the Cerrado domain, except in southern Tocantins state (Figure 4c).

The tree biomass reduction due to fire events is much higher in magnitude more than due to P limitation or inter-annual climate variability (Figure 4d). The greater water availability is related to small



or null fire effect in the Central Amazon rainforest agrees with the fact that Amazonia is naturally
inflammable as well as a gradient towards seasonally dryer climate that increases the intensity and
magnitude of fire effects towards the Cerrado (Figure 4d). The fire effect on TB over the Amazon domain
was 21-24% of the P limitation effect (range for PR and PG cases), while the fire effect on TB over the
Cerrado was more than 250% of the P limitation effects in CV simulations, which is due to quick growth
of grasses after fire occurrence in the latter.
**3.1.2   Influence of climate, fire and phosphorus in the transects**

Results of the ANOVAs and Tukey-Kramer test indicate that the inclusion of CV, limitation by P

(PR and PG) and fire in INLAND led to significantly different averages of NPP, LAI and AGB in the
transition zones. This influence of climate, P and fire are shown separately in Tables 3 to Table 5 and
combined in Table 6.

The effects of climate and P on productivity show that CV reduces the NPP from

0.68 kg-C m$^{-2}$ yr$^{-1}$ to 0.64 kg-C m$^{-2}$ yr$^{-1}$ (Table 3) and the P effect results in NPP decline from
0.71 kg-C m$^{-2}$ yr$^{-1}$ to 0.64 kg-C m$^{-2}$ yr$^{-1}$ (both PR and PG) (Table 4). The fire effect, moreover, has a
positive effect on NPP from 0.66 kg-C m$^{-2}$ yr$^{-1}$ when fire is off to 0.67 kg-C m$^{-2}$ yr$^{-1}$ when fires is on. This
difference, albeit low, is statistically significant (Table 5).

In addition CV and P limitation reduce the LAI$_{total}$ in the canopy (Table 3 and Table 4), increasing

three times LAI$_{lower}$ and decreasing LAI$_{upper}$ (Table 5). The magnitude of fire effect on AGB (46.7%,
Table 5) is greater in relation to the CV (5%, Table 3) and P (14%, Table 4) limitation effects.

Even though CV effects on NPP and AGB for each simulation is not statistically significant, the

effects of fire and P limitation (regardless of phosphorus map) are. Fire effects are significant only for



300 structural variables as AGB, $LAI_{total}$, $LAI_{upper}$ and $LAI_{lower}$. It presents an increase of LAI total of

301 1.52 $m^2$ $m^{-2}$ in CV+PG+F in relation to CV+PG, and of 1.32 $m^2$ $m^{-2}$ in CV+PR+F in relation to CV+PR

302 (Table 6).

### 3.1.3 West-East patterns of AGB in the Amazon-Cerrado transition

304  The model used in this study simulates > 80% of the observed AGB variability in all treatments

305 along the transition area except in T5 (Table 7). It shows that the model is able to capture AGB variability

306 along the transition area, which is relevant when compared to studies that simulate 50% of the observed

307 AGB variability (Senna et al., 2009; Castanho et al., 2013).

308  It is not possible to identify a treatment that best represents AGB in all transects (Table 7). A

309 combined analysis of Table 7 and Figure 5 indicates a general agreement that observed AGB decreases

310 from W to E in T1 to T4, and this is well captured by several configurations of the model, with specific

311 differences among them. Overall, CA and PC configurations, being the least disturbed treatments, yield

312 higher AGB, while the introduction of CV, PG and F reduce the AGB. However, the simulated results

313 may be above or below the observed ones, which suggests that additional local factors are not included

314 in the model.

315  The curves of AGB (Figure 5) show the impact of CV, PG and F along the W-E transition. PG

316 has a high influence on the transition, decreasing the ABG especially in the western part of the transects,

317 where the Amazon vegetation is predominant. This feature is particularly simulated in T3 and T4, where

318 PG decrease the AGB by 2 kg-C $m^{-2}$ in the west pixels of these transects (Figure 5). In T1, T2 and T5,

319 AGB decline is also higher with P limitation when compared to the curves limited only by CV. However,

320 in T1 model simulations tend to underestimate the highest and the lowest AGB extremes, and the absolute



values were always underestimated, despite the improvement in correlation with the inclusion of the fire
component (Table 7).
Fire, however in T2, T3 and T4, is responsible to approach the simulated AGB to the observed
AGB in the eastern pixels into Cerrado domain (Figure 5). In T5 these relations are similar, with climate
presenting less influence on AGB decrease than P, and fire appears mainly as a reducer factor.
**3.2    Simulated composition of vegetation**
Most of the pixels in CA show very robust simulations, with more than 90% of the same vegetation
cover in the simulated last 10 years (Figure 6a-c and 6g-i). A larger number of pixels with transitional
vegetation were simulated in CV (Figure 6d-f and and 6j-l). An even higher variability in CV compared
to CA simulations was observed when we added the effects of P limitation and fire (Figure 6a and 6j-l).
The vegetation composition in all P limitation scenarios for CA simulations resulted in robust
simulations for nearly all pixels, except for the north of Cerrado domain (Figures 6a, 6b and 6c). The
CA+PC and CA+PR simulations had the same vegetation composition, while CA+PG replaced the
deciduous forest by evergreen forest in the central Cerrado region, around 8°S 46°W (Figures 6A, 6B and
6C). This behavior might be related to the higher $P_{total}$ values in PG than PR and PC for the Cerrado region
(Figure S1). Cerrado was better represented in CV+PC, CV+PR and CV+PG than in the same CA
combinations (Figure 6). The occurrence of forested areas in central Cerrado decreased in CV
combinations, these being replaced by the savanna or grassland vegetation class.
When the effect of fire was added to CA simulations, the model simulated an increase in the
uncertainty on the vegetation cover classification in the Cerrado region. The effect of fire reduced the
presence of deciduous forest in central Cerrado biome as well as in CA+PC, and the vegetation was



replaced by evergreen forest and savanna in CA+PC+F (Figures 6G, 6H, 6I). In CV simulations, fire
effect results in the replacement of the deciduous and perennial forest by savanna and grasses in all central
Cerrado region (Figures 6J, 6K and 6L).

For all combinations used, transitional forest areas in the northern and southwestern Cerrado

biome are not adequately represented. With >90% of concordance, INLAND assigns the existence of
tropical evergreen forest rather than deciduous forest in some pixels in the north of the transition, and the
existence of tropical evergreen forest rather than savanna in the southwest, indicating difficulty to
simulate transitional vegetation in these regions.
**4    Discussion**

The inclusion of CV, PR and PG and fire in INLAND showed significant influences on the

simulated vegetation structure and dynamics in the Amazon-Cerrado border (Figure 4 and Table 6),
suggesting that these factors play key role on vegetation structure in the forest-savanna border and can
improve the simulated representation of the current contact zone between these biomes. This is broadly
consistent with the literature that investigated causes of savanna existence in the real world (Hoffmann et
al., 2012; Dantas et al., 2013; Lehmann et al., 2014). In this study, the spatial analysis and the Tukey-
Kramer test (TK) show a difference in magnitude among these factors in vegetation, with fire occurrence
and P limitation being stronger than inter-annual climate variability along the transects (Figure 4).

The spatial analysis showed that CV declines AGB predominantly in eastern Amazonia (Figure

4a). Climate of this region is intensely affected by El Niño–Southern Oscillation (ENSO), which could
reduce precipitation by 50%, placing the vegetation under intense water stress (Botta and Foley, 2002;
Foley et al. 2002; Marengo et al., 2004; Andreoli et al., 2013; Hilker et al., 2014). This reduction in



rainfall in dry years brings in direct changes in carbon flux (NPP) and stocks in leaves and wood, leading
to changes in vegetation structure. In addition to inter-annual changes in the rainfall, inter-annual
variability in other climate variables in CV also affect AGB, as average, maximum and minimum
temperature, as well as wind speed and specific humidity, and influence photosynthesis on the model both
directly (through Collatz and Farquhar equations) and indirectly (e.g. through evapotranspiration). Our
results showed significant differences for most part of the biomes, except central Amazonia (Figure 4a),
where CV and precipitation seasonality have been pointed as secondary effects on vegetation (Restrepo-
Coupe et al., 2013), since there is no shortage of water availability during the dry season.

Along the Cerrado, lower water availability in some years in CV affects tree biomass, although

that vegetation is predominantly grassy-herbaceous. The AGB decline is significant for most part of the
simulated Cerrado domain (Figure 4a) and average values could represent half the amount of typical tree
biomass in this biome. This reduction in AGB reflects INLAND's ability to simulate similar Cerrado
conditions and expose the few trees to high water stress.

Throughout the transects, however, no significant difference was found for average AGB between

CV+PC and CA+PC by TK at $p<0.05$ (Table 6). On the other hand, when we analyzed the influence of
CV for the same pixels, but using all simulations (Table 3), regardless of P limitation and fire occurrences,
the results showed that the decrease in AGB by 0.38 kg-C m$^{-2}$ (5.7%) is statistically significant along the
transition.

P limitation effect was statistically significant for PR and PG along all the Amazon domain and

the main differences between these simulations were the spatial patterns of tree AGB decrease (Figure 4b
and Figure 4c). We cannot affirm which of these databases is better because they are the results of



different methodologies and observations (Quesada et al., 2009; Yang et al., 2014). However, PG showed
a higher AGB decrease in central Amazonia, northeastern Pará and northeastern Mato Grosso state,
indicating that in these areas the P limitation is higher. This result does not corroborate the northwest-
southeast AGB gradient found in the Amazon basin, which showed a higher productivity in the west
where soils are more fertile than those found in the southeast (Aragão et al., 2009; Saatchi et al., 2007;
Nunes et al., 2012; Lee et al., 2013). On the other hand, PR AGB agrees with the northwest-southeast
gradient, presenting less limitation in the soils of central Amazonia with declines in AGB mainly in the
southeastern part of the rainforest (between Pará and northeastern Mato Grosso states) (Figure 4b).
In Cerrado, P limitation also influenced vegetation (Figure 4c) and presented statistically
significant differences when compared to CV+PC. In this biome, as well as in the Amazon, tree abundance
richness and diversity have been generally associated with increases of soil fertility (Long et al., 2012;
Vourtilis et al., 2013), highlighting the importance of P in the composition and maintenance of vegetation,
especially in transition areas.
Compared to the Amazon domain, the magnitude of effects of P limitation is lower in the Cerrado.
However, few pixels in PR that have P limitation showed a significant decrease in arboreal AGB (Figure
4b), while in PG, we found reduction of AGB for most of the Cerrado domain, except only for the southern
Tocantins state (Figure 4c). Despite the differences in spatial patterns, there was no statistically significant
differences between PR and PG within the transects (Table 4 and Table 6).
The spatial difference between PG and PR showed that PG is lower than PR in the western
Amazonia, and higher in northern Amazonia. Moreover, PG have low P values in south of the transition
compared to PR, while in Cerrado domain P values ranged between 120 to 200 mg kg$^{-1}$ (Figure S1).



Although the PR dataset includes every known P data collected in the region, these differences reinforce
the need to improve the data of $P_{total}$ in the soils of the Amazon and Cerrado/Amazon transition domains.
Currently, $P_{total}$ data in Cerrado is scarce, and make unfeasible to establish a proxy similar to Castanho et
al. (2013), which was specific for the Amazon.

To our knowledge, the most part of the Dynamic Global Vegetation Models (DGVMs) do not

consider the complete phosphorus cycle, despite the importance of nutrient cycling for AGB maintenance
and tropical vegetation dynamics in dystrophic soils. For example, nutrient cycling in the
Amazon/Cerrado transition is closely related to the hyper-dynamic turnover of the AGB (Valadão et al.
2016), in which some key species might also be crucial to the hyper-cycling of nutrients through which
vegetation sustain the constant input of nutrients, including large annual amounts of available P (Oliveira
et al. 2017).

The decrease in tree AGB occasioned by P limitation can contribute to a decrease in litter

production and consequently could affect nutrient cycling in tropical ecosystems. According to Oliveira
et al. (2017), the litter produced by vegetation corresponds to the main return route to the available fraction
of P for plants, especially in the transition areas, where $P_{available}$ in the soil is very low. In our model,
however, P acts directly in the photosynthesis limitation through $V_{max}$ and cannot be reabsorbed by the
roots. Thus, the litter produced in vegetation contribute only to dry matter and fire occurrence increase.
In nature, the litter affected by fire occurrence volatilizes the small amount of P available to plants,
increasing the nutrient losses of the ecosystem. Despite this simplified representation in INLAND, it can
represent the P influence on woody AGB in the Amazon and Cerrado.



The fire occurrence is an important factor controlling the AGB dynamics in the Cerrado or in the
transition vegetation (Silvério et al., 2013; Couto-Santos et al., 2014; Balch et al., 2015), which this study
clearly replicates, showing statistically significant influences when compared to control simulations
(Figure 4d and Table 5). In the transition, the fire effect may reduce average AGB by 50%, which under
climate change or deforestation conditions may lead to an even stronger change in the vegetation structure
and dynamic. In the Cerrado domain, the simulated fire effect implies in significant increases of shrubs
and herbaceous vegetation and decreases of the arboreal component. In nature, however, the Cerrado is
relatively resilient to fire depending on the velocity, intensity and duration of the burning (Rezende et al.,
2005; Elias et al., 2013 Reis et al., 2015). The adaptive morphological nature and the low nutrient
requirement of vegetation allow Cerrado the capacity to rapidly restore the vegetation after fire occurrence
(Hoffmann et al., 2005; Hoffmann et al., 2012). In our model, the restoration of vegetation after fire
occurrence is exclusively due to the canopy opening and consequently more luminosity penetration into
lower canopy.
This study shows an improvement in the correlations between simulated and observed AGB when
compared to previous modeling studies, regardless of treatment, with correlation coefficients usually
above 0.80 for the transects, except for T5, for which the correlation coefficient value is usually below
0.5 (Table 7). Senna et al. (2009) found 0.20 as maximum correlation coefficient between simulated and
observed ABG while Castanho et al. (2013) showed 0.80 for Amazonia domain. From Figure 5, it is clear
that CV, F and P limitation in the transition zone reduce the AGB, approaching the simulated to the
observed data, and play important roles in the simulations, but the only inclusion of these effects is still
insufficient to represent the actual vegetation structure in the Amazon-Cerrado border (Figure 6L). In our



interpretation, this means that other important factors are still missing from the simulation, especially in
T5, where soils are rocky and shallow. A better spatial representation of soil physical properties, including
shallow rocky soils, as well as spatially varying physiological parameterizations of the vegetation such as
carbon allocation, deciduousness of vegetation, and residence time are probably needed to improve the
simulations, in particular in the northern and southern extremes of the border (T1 and T5).

In addition, literature shows that in the transition area, soils are very different than Amazon soils,

and that essential proprieties for modeling are peculiar (Silva et al., 2006; Vourlitis et al., 2013; Dias et
al. 2015). For example, Dias et al. (2015) recently showed that the pedological functions normally used
by DGVMs may underestimate the saturated hydraulic conductivity ($K_s$) by >99%, transforming a well-
drained soil with $K_s = 1.5.10^{-4}$ m.s$^{-1}$ (540 mm.h$^{-1}$) in reality into an impervious brick with
$K_s = 3.3.10^{-7}$ m.s$^{-1}$ (1.2 mm.h$^{-1}$) in the model.

For all transects, the AGB curves have similar patterns (Figure 5); the smaller difference is

observed between CA+PC and CV+PG curves, while the larger difference is when fire is present. The
effect of P limitation appears as an effect of intermediate magnitude, reducing the AGB by more than the
effect of inter-annual climate variability. In the east, it is observed that there is little or no difference
among AGB simulated by CA+PC, CV+PC and CV+PG, revealing that inter-annual climate variability
and P have smaller influence in the AGB. However, in the east of T2, T3 and T4, fire is the factor that
adjusts the simulated to the observed data (Figure 5), differently than the grid points in the West, where
CV+PG is a better proxy between observed and simulated data.

Such conditions are interesting because they reflect the different mechanisms that regulate the

structure of these ecosystems and probably the phytophysiognomies distribution. For example, P





limitation seems to be the factor that improves simulated AGB in regions where the predominant
vegetation type is the tropical rainforest. Fire, on the other hand, improves the AGB in grid points where
the Cerrado occurs. Moreover, important factors such as productivity partitioning into leaves, roots and
wood carbon pools are assumed to be fixed in space and time within a given PFT, neglecting the natural
capacity of transitional forests to adapt itself and to adjust their metabolism to local environmental
conditions (Senna et al., 2009). In years of severe drought, transitional forests could prioritize the stock
of carbon to fine roots instead of the basal increment to maximize access to available water, or make
hydraulic redistribution to maintain the greenness and photosynthesis rates. Brando et al. (2008) found
high sensitivity in carbon allocation for eastern Amazon basin trees, which reduced wood production by
13-60% in response to an artificial drought. Although in INLAND soil moisture can reduce the
photosynthetic rates during the months of lower rainfall, it does not dynamically change the allocation
rates, exposing the PFTs in these areas to severe water stress and underestimating the AGB, such as in
the west of T1 (Figure 5a).

T2, T3 and T4, located in the central part of the Amazon-Cerrado transition, showed the highest

average correlations between observed and simulated data (Table 7). For these transects, INLAND seems
to be able to capture the high variability of AGB gradient.

At T5, located at the south of the transition, the average correlations were low for all treatments,

indicating that INLAND has difficulty to represent the AGB gradient there (Table 7). However, it captures
the lower AGB as compared to the northern ones. In this region, the vegetation is characterized by a wide
diversity of physiognomies, which varies with other preponderant factors, such as lithology, soil depth,
topography and fertility. The observed data also showed high AGB variability, indicating that there are





changes in the vegetation structure, featuring medium-sized and small vegetation types on different soil
types. In INLAND, however, features such as lithology and water-table depth are not considered due to
the complexity of its representation on the large scale, limiting the representation of a heterogeneous
environment throughout the transition.

Different patterns of vegetation distribution along the Amazon-Cerrado border exist and are

influenced not only by inter-annual climate variability, P limitation, and fire, but also by the
ecophysiological parameters, which may have different behavior according to the environmental
conditions and soil proprieties. Obtaining these parameters is a challenge to the scientific community
once the field measurements are difficult due to the extension of the transition area. More observed data
are needed to establish and implement the plasticity of the fixed parameters such as carbon coefficients
allocation, residence time, dependence of the deciduousness on P, among others.

Another point to discuss is that the model simulates, in a few pixels in southeastern Cerrado, very

robust simulations of the presence of savanna and grassland even in the absence of fire (Figure 6A-F and
6a-f). This is, in our view, a result of the intense water and heat stress in this region. In the Brazilian
Cerrado, the high temperatures ($> 35\ °C$) combined to the dry season duration (as long as 6 months with
little or no rain) exposes the vegetation to a severely stressed situation, so that a low biomass, low LAI
vegetation may exist without the need of a frequent disturbance.

## 5    Conclusions

This is the first study that uses modeling to assess the influence of inter-annual climate variability,

fire occurrence and phosphorus limitation to represent the Amazon-Cerrado border. This study shows





that, although the model forced by a climatological database is able to simulate basic characteristics of
the Amazon-Cerrado transition, the addition of factors such as inter-annual climate variability,
phosphorus limitation and fire gradually improves simulated vegetation types. These effects are not
homogeneous along the latitudinal/longitudinal gradient, which makes the adequate simulation of
biomass challenging in some places along the transition. Our work shows that fire is in the main
determinant factor of the vegetation changes along the transition. The nutrient limitation is second in
magnitude, stronger than the effect of inter-annual climate variability.

Overall, although INLAND typically simulates more than 80% of the variability of biomass in the

transition zone, in many places the biomass is clearly not well simulated. Situations for clearly wet or
markedly dry climate conditions were well simulated, but the simulations are generally poor for
transitional areas where the environment selected physiognomies that have an intermediate behavior, as
is the case of the transitional forests in northern Tocantins and Mato Grosso.

The lack of field parameters measured in the transition zone is still a major limitation to improve

the DGVMs. Spatially explicit carbon allocation strategies, mortality rates, physiological and structural
parameters are necessary to establish numerical relationships between the environment and the vegetation
dynamic models to make them able to correctly simulate current patterns and future changes in vegetation
considering future climate change. In addition, it is also needed to include not only the spatial variability,
but also temporal variability in physiological parameters of vegetation, allowing a more realistic
simulation of the vegetation-climate relationship. Finally, our results reinforce the importance and need
of the DGVMs to incorporate the nutrient limitation and fire occurrence to simulate the actual Amazon-
Cerrado border position.


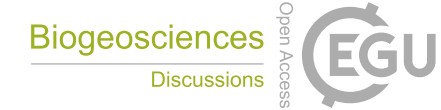


## 6 Acknowledgements

We gratefully thank FAPEMIG and CAPES (Brazil) for their financial support. Atul K Jain is funded by the U.S. National Science Foundation (NSF-AGS- 12-43071).

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



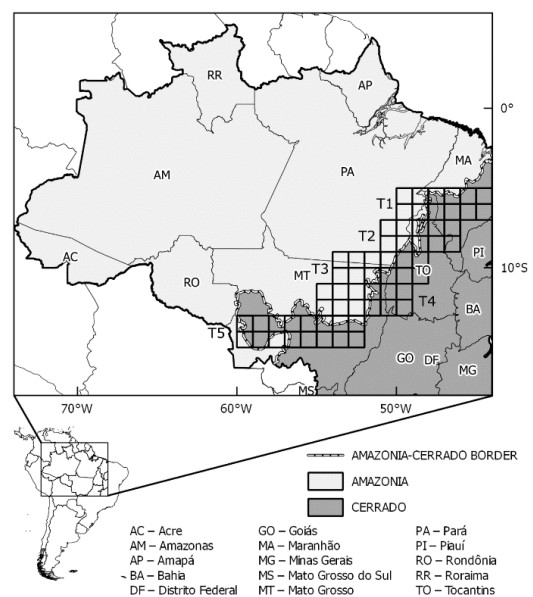


**Figure 1.** Delimitation of the study area Amazonia (in light gray) and Cerrado (in dark gray) (IBGE,

2004), and the location of five transects used in this work (from T1 to T5). The dashed line represents the

border between biomes.



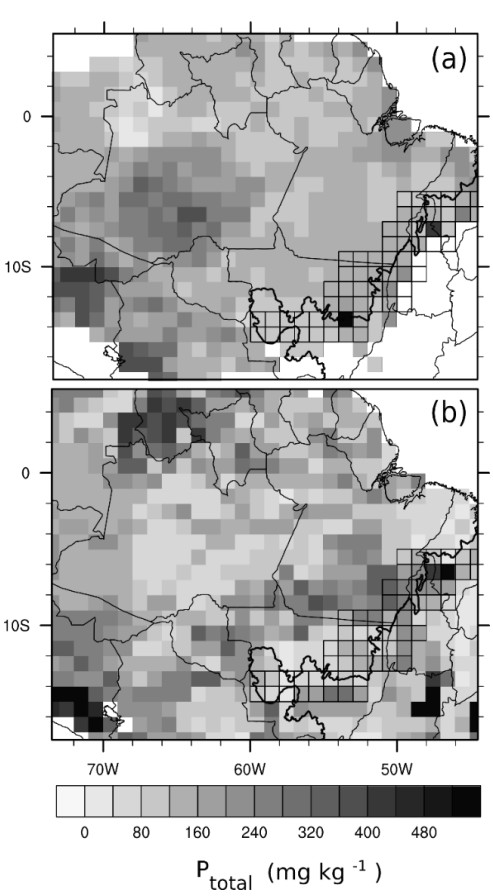


**Figure 2.** (a) Map of regional total P in the soil (PR), (b) Map of global total P in the soil (Yang et al.,

2013) (PG).



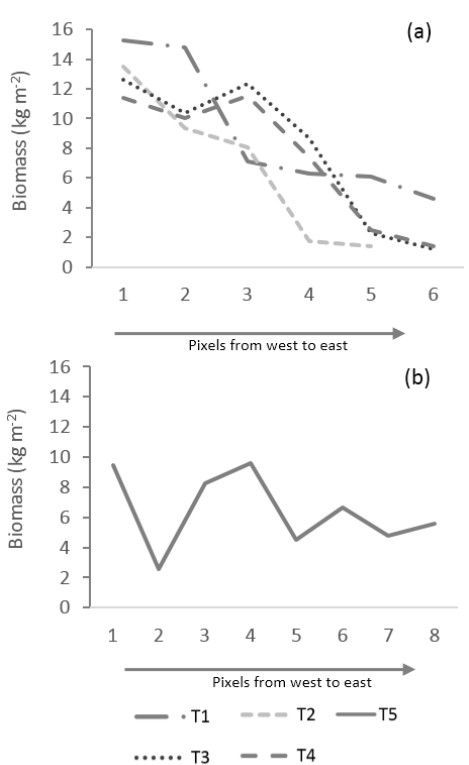


**Figure 3.** Variations of AGB in pixels from West to East in the Amazonia-Cerrado transition for

transects T1, T2, T3 and T4 (a), and T5 (b).



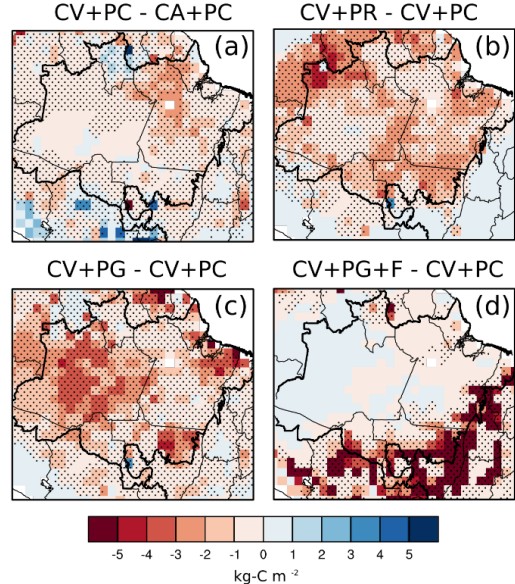


**Figure 4.** Effects of inter-annual climate variability (a), Regional P limitation (b), Global P limitation (c),

and fire (d) on AGB. The hatched areas indicate that the variables are significantly different compared to

the control simulation at the level of 95% according to the t-test. The thick black line is the geographical

limits of the biomes.





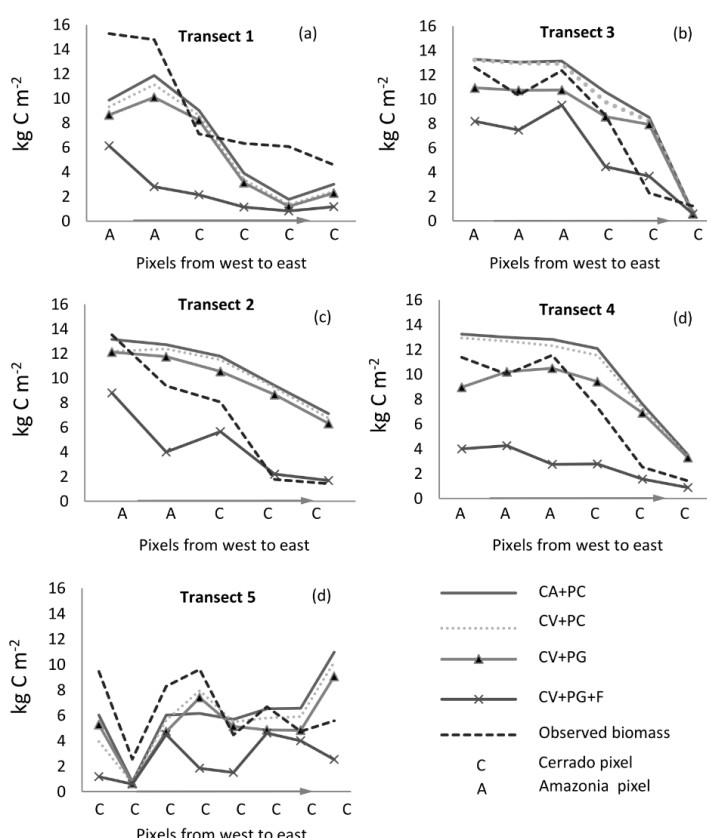

**Figure 5.** Longitudinal AGB gradient in Amazonia-Cerrado transition simulated for T1 to T5 considering

different combinations: observed data; seasonal climate control simulation (CA+PC); inter-annual

climate variability (CV+PC); inter-annual climate variability + global P limitation (CV+PG); and inter-

annual climate variability + P + fire occurrence (CV+PG+F).



813

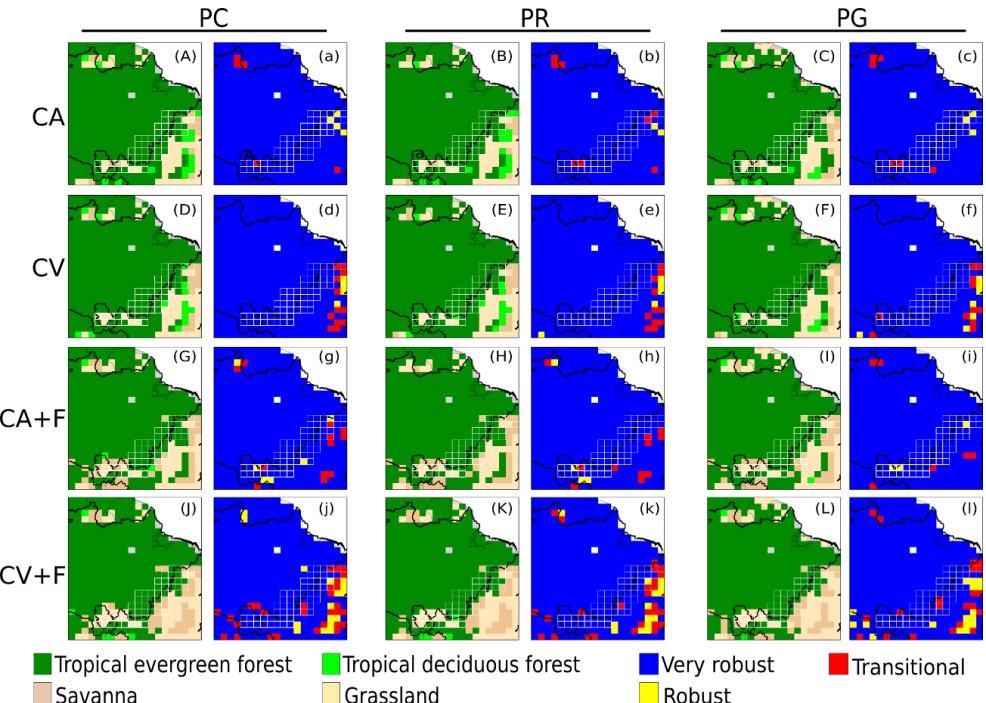

814

**Figure 6.** Results for the dominant vegetation cover simulated by INLAND for the different treatments

(A-L) and a metric of variability of results (a-l). Simulations are considered very robust if the dominant

vegetation agrees on 9-10 of the last 10 years of simulation, robust if it agrees on 7-8 years, and

transitional if on 6 or fewer years.







**Table 1.** Simulations with different scenarios evaluated by INLAND model in Amazonia-Cerrado
transition. CA, climatological average, 1961-1990; CV, monthly climate data, 1948-2008; the nutrient
limitation on $V_{max}$ - PC, no P limitation ($V_{max}$ = 65 μmol-$CO_2$ $m^{-2}$ $s^{-1}$); PR, regional P limitation; PG,
global P limitation).

| | | | $V_{max}$ | | |
|---------|----------|----------|----------|----------|----------|
| Climate | $CO_2$ | Fire (F) | PC | PR | PG |
| CA | Variable | Off | CA+PC | CA+PR | CA+PG |
| CA | Variable | On | CA+PC+F | CA+PR+F | CA+PG+F |
| CV | Variable | Off | CV+PC | CV+PR | CV+PG |
| CV | Variable | On | CV+PC+F | CV+PR+F | CV+PG+F |





**Table 2.** Individual and combined effects for each simulation in Amazonia-Cerrado transition. CA,
climatological seasonal average, 1961-1990; CV, monthly climate data, 1948-2008; the nutrient limitation
on $V_{max}$ - PC, no P limitation ($V_{max} = 65$ μmolCO$_2$ m$^{-2}$ s$^{-1}$); PR, regional P limitation; PG, global P
limitation)

| Climate (C) | Phosphorus (P) | Fire (F) |
| --- | --- | --- |
| (CV+PC)-(CA+PC) | (CA+PR)-(CA+PC) | (CA+PC+F)-(CA+PC) |
| (CV+PR)-(CA+PR) | (CV+PR)-(CV+PC) | (CV+PC+F)-(CV+PC) |
| (CV+PG)-(CA+PG) | (CA+PG)-(CA+PC) | (CA+PR+F)-(CA+PR) |
| | (CV+PG)-(CV+PC) | (CV+PR+F)-(CV+PR) |
| | | (CA+PG+F)-(CA+PG) |
| | | (CV+PG+F)-(CV+PG) |







**Table 3.** Summary of average NPP, LAI and AGB for the Amazonia-Cerrado transition at the transects
domains, considering all simulations with CA and CV regardless of fire presence or P limitation. The
results of a one-way ANOVA are also shown, including the $F$ statistic, and p value. Values within each
column followed by a different letter are significantly different ($p < 0.05$) according to the Tukey–Kramer
test (n=1860: 31 pixels x 10 years x $n_{simulation/2}$).

| Group 1 | NPP | | $LAI_{total}$ | | $LAI_{lower}$ | | $LAI_{upper}$ | | AGB | |
|---|---|---|---|---|---|---|---|---|---|---|
| | kg-C m$^{-2}$ yr$^{-1}$ | | m$^2$ m$^{-2}$ | | m$^2$ m$^{-2}$ | | m$^2$ m$^{-2}$ | | kg-C m$^{-2}$ | |
| CA | 0.68 | a | 7.47 | a | 1.98 | a | 5.49 | a | 6.68 | a |
| CV | 0.64 | b | 7.15 | b | 2.11 | a | 5.04 | b | 6.30 | b |
| $F$ | 40.2 | | 57.2 | | 2.96 | | 36.0 | | 11.3 | |
| $p$ | <0.001 | | <0.001 | | ns | | <0.01 | | <0.001 | |





**Table 4.** Summary of average NPP, LAI and AGB for the transition at the transects domains, considering
different P limitation, regardless of climate and fire presence. The results of a one-way ANOVA are also
shown, including the $F$ statistic, and p value. Values within each column followed by a different letter are
significantly different ($p < 0.05$) according to the Tukey–Kramer test (n=1240: 31 pixels x 10 years x
$n_{simulation/3}$).

| Group 2 | NPP | | LAI$_{total}$ | | LAI$_{lower}$ | | LAI$_{upper}$ | | AGB | |
|---|---|---|---|---|---|---|---|---|---|---|
| | kg-C m$^{-2}$ yr$^{-1}$ | | m$^2$ m$^{-2}$ | | m$^2$ m$^{-2}$ | | m$^2$ m$^{-2}$ | | kg-C m$^{-2}$ | |
| PC | 0.71 | a | 7.64 | a | 1.84 | b | 5.80 | a | 7.15 | a |
| PR | 0.64 | b | 7.15 | b | 2.19 | a | 4.95 | b | 6.20 | b |
| PG | 0.64 | b | 7.14 | b | 2.10 | a | 5.04 | b | 6.12 | b |
| $F_{2.99}$ | 62.8 | | 61.0 | | 8.75 | | 53.5 | | 33.6 | |
| $p$ | <0.001 | | <0.001 | | <0.01 | | <0.01 | | <0.001 | |





**Table 5.** Summary of average NPP, LAI and AGB for the transition at the transects domains, considering
presence or absence of fire. The results of a one-way ANOVA are also shown, including the *F* statistic,
and p value. Values within each column followed by a different letter are significantly different ($p < 0.05$)
according to the Tukey–Kramer test (n=1860: 31 pixels x 10 years x $n_{simulation/2}$).

| Group 3 | NPP | | $LAI_{total}$ | | $LAI_{lower}$ | | $LAI_{upper}$ | | AGB | |
|---|---|---|---|---|---|---|---|---|---|---|
| | kg-C m$^{-2}$ yr$^{-1}$ | | m$^2$ m$^{-2}$ | | m$^2$ m$^{-2}$ | | m$^2$ m$^{-2}$ | | kg-C m$^{-2}$ | |
| Fire OFF | 0.66 | a | 6.72 | b | 0.88 | b | 5.84 | a | 8.47 | b |
| Fire ON | 0.67 | b | 7.90 | a | 3.21 | a | 4.69 | b | 4.51 | a |
| $F_{3.84}$ | 8.28 | | 937 | | 1459 | | 249 | | 1719 | |
| *p* | *<0.005* | | *<0.001* | | *<0.01* | | *<0.01* | | *<0.001* | |





**Table 6.** Summary of average NPP, LAI and AGB for the transition at the transects domains, considering
all factor combinations. The results of a one-way ANOVA are also shown, including the *F* statistic, and
p value. Values within each column followed by a different letter are significantly different (p < 0.05)
according to the Tukey–Kramer test (n=310: 31 pixels x 10 years).

| | NPP $kg\text{-}C\ m^{-2}\ yr^{-1}$ | | $LAI_{total}$ $m^2\ m^{-2}$ | | $LAI_{lower}$ $m^2\ m^{-2}$ | | $LAI_{upper}$ $m^2\ m^{-2}$ | | AGB $kg\text{-}C\ m^{-2}$ | |
|---|---|---|---|---|---|---|---|---|---|---|
| CV+PC | 0.69 | bcd | 6.96 | d | 0.84 | e | 6.48 | a | 9.01 | ab |
| CV+PG | 0.61 | f | 6.24 | f | 0.85 | e | 5.60 | bc | 7.91 | c |
| CV+PR | 0.62 | f | 6.33 | f | 0.85 | e | 5.74 | bc | 8.04 | c |
| CV+PC+F | 0.69 | abc | 7.92 | b | 2.91 | cd | 4.61 | ef | 4.89 | de |
| CV+PG+F | 0.63 | ef | 7.76 | b | 3.73 | a | 5.81 | bc | 3.91 | f |
| CV+PR+F | 0.63 | ef | 7.65 | bc | 3.47 | ab | 4.69 | ef | 4.02 | f |
| CA+PC | 0.72 | ab | 7.39 | c | 0.91 | e | 6.12 | ab | 9.31 | a |
| CA+PG | 0.64 | def | 6.64 | e | 0.91 | e | 5.40 | cd | 8.22 | c |
| CA+PR | 0.65 | cdef | 6.72 | de | 0.91 | e | 5.49 | cd | 8.31 | bc |
| CA+PC+F | 0.74 | a | 8.29 | a | 2.69 | d | 5.02 | de | 5.40 | d |
| CA+PG+F | 0.67 | cde | 7.90 | b | 3.29 | abc | 4.04 | g | 4.45 | ef |
| CA+PR+F | 0.67 | cde | 7.88 | b | 3.19 | bc | 4.18 | fg | 4.42 | ef |
| *F* | *16.2* | | *115* | | *140* | | *38.1* | | *172* | |
| *p* | *<0.001* | | *<0.001* | | *<0.01* | | *<0.01* | | *<0.001* | |




**Table 7**. Correlation coefficients of AGB simulated by INLAND and field estimates (n= 310: 31 pixels
x 10 years).

|          | T1    | T2    | T3    | T4    | T5    | All transects |
|----------|-------|-------|-------|-------|-------|---------------|
| CA+PC    | 0.843 | 0.928 | 0.886 | 0.937 | 0.337 | 0.786         |
| CV+PC    | 0.838 | 0.884 | 0.890 | 0.939 | 0.355 | 0.781         |
| CA+PR    | 0.793 | 0.848 | 0.830 | 0.911 | 0.399 | 0.756         |
| CV+PR    | 0.795 | 0.793 | 0.832 | 0.907 | 0.527 | 0.771         |
| CA+PG    | 0.814 | 0.951 | 0.838 | 0.889 | 0.388 | 0.776         |
| CV+PG    | 0.825 | 0.922 | 0.840 | 0.879 | 0.496 | 0.792         |
| CA+PC+F  | 0.988 | 0.987 | 0.977 | 0.892 | 0.133 | 0.795         |
| CV+PC+F  | 0.976 | 0.947 | 0.933 | 0.908 | 0.187 | 0.790         |
| CA+PR+F  | 0.842 | 0.805 | 0.981 | 0.808 | 0.561 | 0.799         |
| CV+PR+F  | 0.925 | 0.804 | 0.927 | 0.808 | 0.319 | 0.757         |
| CA+PG+F  | 0.844 | 0.961 | 0.980 | 0.830 | 0.430 | 0.809         |
| CV+PG+F  | 0.845 | 0.932 | 0.931 | 0.881 | 0.177 | 0.753         |
| CA avg   | 0.854 | 0.913 | 0.915 | 0.878 | 0.375 | 0.787         |
| CV avg   | 0.867 | 0.880 | 0.892 | 0.887 | 0.344 | 0.774         |
