# Peer review of "Influence of climate variability, fire and phosphorus limitation on the"

_Biogeosciences, 2017_

## Referee Comment (RC1) · Anonymous Referee #1 · 11 Jul 2017

General Comments:

Reviewer summary: The manuscript presents results from the DGVM INLAND evaluating the effect of climate variability, fire and phosphorus limitation on vegetation structure and dynamics in the transitional zone between the forested Amazon and grassland savanna of the Cerrado. Changes in net primary production, aboveground biomass, and leaf area index are assessed between simulations, and simulated aboveground biomass is compared to observations. Transects along the Amazon-Cerrado transition zone are analyzed in subsets, as well as the region as a whole. Inclusion of climate

variability, fire and maps phosphorus limitation improves simulation of vegetation structure across the region and for four of five transects. The cerrado transect has the lowest correlation to observations. Fire has the strongest impact on vegetation characteristics, followed by phosphorous limitation. Overall, INLAND with these included factors appears effective at simulating vegetation across the region, but regional deficiencies show that more improvements can be made.

Article contribution and overall impact: This study highlights the need for improved simulation of vegetation in a key forest-savanna transition zone. A shift in vegetation in this region has the capacity to impact the cycling of water and nutrients as well as energy fluxes beyond the area of forest-savanna transition. Uncertainty in future climate and fire behavior as well as the feedback between these factors make the vegetation state of this region difficult to predict. The manuscript does a good job of presenting the challenges of simulating vegetation in this transitional zone where climate, nutrients and fire are essential contributors to vegetation state. The inclusion of phosphorous limitation in the simulations for the region is an important addition to the evaluation of vegetation state. The discussion would benefit from a more detailed description of the fire model and fire activity during simulation. The concluding recommendations for future work also need to be clarified. Discussion of the importance of vegetation size structure and how this is or is not represented in INLAND should also be added. A key component of the mortality of woody vegetation to fire is its size at the time of fire and the ability to accumulate size between fires. This is central to the work of many of W. Hoffman's papers in the region (Hoffman et al 2003, Hoffman et al 2009, Hoffman et al 2012). Please add discussion of size structure to the manuscript.

Detailed comments:

Page 6 line 101-107: Add the 1x1 degree grid size to this section.

Page 9 line 159-164: Provide more detail concerning fire model. Is the fire ignition probability the same in every pixel, or is there spatial variability? If there is spatial

variability what drives this parameterization? Does flammability vary by PFT?

Page 16 line 316: define "transition" here and throughout manuscript. The reference is not always clear. Most often it appears to be Amazon-Cerrado transition or forest-savanna transition, if that is correct please add the extra detail here and throughout manuscript (Pg 20 line 403, pg 21 line 419, pg 22 line 426,428,443, pg 23 line 451)

Page 17 line 323-325: Update sentence to "responsible for altering the simulated AGB to approach the observed AGB" or some variant. Current sentence structure is unclear.

Page 18 line 345-349: What observational data set is this being compared to for current vegetation state? Is this a by pixel comparison of the same grid size?

Page 18 line 359-364: Do the climate datasets used in simulation include reduced rainfall and ENSO effects? Explain this further.

Page 21 line 409: Update to "for the most part Dynamic Vegetation Models"

Page 22 line 435-437: Add more detail clarifying how the INLAND model differs from reality. Is it able to simulate rapid restoration following fire? If not, what would need to be added to the model's fire or vegetation characteristics?

Page 22 line 444: Update to "but the inclusion of these effects"

Page 22 line 442-445: This needs more explanation. Is the vegetation simulation insufficient because of the presence of transitional and robust pixels in the cerrado in fig 5? Or is this because of comparisons to observed data of vegetation in the cerrado?

Page 23 line 449: what is meant by residence time?

Page 24 line 477-479: Explain this in more detail: "It does not dynamically change the allocation"

Page 25 line 495-496: Reword this sentence. The meaning is not clear.

Page 26 line 522-525: Inclusion of spatially explicit parameters may or may not improve

DGVM simulation. This assumes that the existing processes are accurate, and that it is merely parameters. Provide more discussion of this possibility, or re-word this section.

Page 26 line 525-527: What is meant by temporal variability? Size structure?

---

## Referee Comment (RC2) · Anonymous Referee #2 · 23 Jul 2017

This work uses the vegetation model INLAND to evaluate the individual and combined effects of the climate variability, the fire and the Phosphorus (P) limitation on the Brazilian ecosystem. The changes on the NPP, ] and AGB were evaluated in relation to 12 climate simulations. The AGB was also evaluated in function of observed data. In addition to climate variability, this work shows the importance of considering the soil nutrient limitation as well as the disturbances caused by the biomass burning in the study of vegetation dynamics. It is also presented some deficiencies of the DGVMs and the databases used to feed the INLAND model. Understanding the mechanisms

that affect the vegetation and the efforts to improve numerical models in order to simulate such effects is of paramount important to the scientific progress. Therefore, this study is of great relevance and, in my opinion, it is suitable for publication in the BG. However, I have some recommendations and doubts that I would like to see being clarified before publication.

**Specific comments**

1. L55-L60: observing the Figure 6d (CV+PC), all the Amazon region became "very robust", so we can assume that the simulation that considers only the climatic effect didn't indicate the "savannization of the Amazon", in other words, the results obtained in this study don't agree with the mentioned works. Can you comment on this?

2. L140: "values smaller than 0.8 $m^2m^{-2}$ characterize a grassland vegetation type" – Grassland can have LAI values much higher than 0.8 $m^2m^{-2}$. Darvishzadeh et al., 2008 found out grassland's average values of 2.76 m2m-2 and maximum value of 7.34 $m^2m^{-2}$. Please check if the INLAND really utilizes this threshold of LAI to define grassland.

3. L85-L87: According to Oliveira et al. (*in press*), the weather also has influence in the nutrients. Then, the climate change's effect cannot be higher due to the indirect effects in the nutrients? Can you comment on this?

4. L157-L158: How are the other PFTs affected by the availability of P?

5. L262: I didn't understand where the 8.7% came from. Could you make it clearer?

6. L339-L344: It can be seen in Figure 6 large differences between $CA + F$ (Line 3) and $CV + F$ (Line 4). However, the differences between $CA$ (Line 1) and

$CA + F$ (Line 3) and the ones between $PC$ (Column 1), $PR$ (Column 2) and $PG$ (Column 3) are not very significant. Thus, the climatic variability is dominant when considering the three effects. Probably, if a fifth map showing $CA + PG + F - CA + PG$ is constructed in Figure 4, it will be quite distinct from Figure 4d. Therefore, it should be exposed more clearly how it came up to the conclusion described in L513-515.

7. L342: I think it is unlikely that an area with "deciduous forest" will turn into "evergreen forest" after being consumed by fire. Please comment if this is possible or if it is a model deficiency.

**Technical corrections**

1. L22: "1960 – 1990" → "1961 – 1990", as described in L204.

2. L23: "two regional datasets" → "two datasets".

3. L62: "particularly the P limitation." → "particularly the Phosphorus (P) limitation."

4. L72: "Phosphorus (P) is a" → "P is a" or "Phosphorus is a".

5. L103: Transects 1 and 2 are more related to "Cerrado" than "Amazon", as shown in Figures 1, 2 and 5. Please rewrite this sentence.

6. L105: "Transect 1 (T1, 43°-49°W; 5°-7°S)" → "Transect 1 (T1, 44°-50°W; 5°-7°S)".

7. L107: "Transect 5 (T5, 53°-61° W; 13°-15° S)" → "Transect 5 (T5, 52°-60° W; 13°-15° S)".

8. L138: "annual mean $LAI_{upper}$ above" → "annual mean $LAI_{upper}$ below"

9. L173: "We used the P-mehlich-1" → "We used the Pmehlich-1"

10. L176: "resulting in 12 additional pixels" – Wouldn't it be 6?

11. L176: "pixels with observed total P content" → "pixels without observed total P content"

12. L186: "Above-Ground AGB (AGB) database" → "Above-Ground Biomass (AGB) database"

13. L194-L196: There are two pixels for each longitude in each transects. Do the Figures 3 and 5 show the mean of the two pixels, or only the upper or the lower one?

14. L212: Remove the phrase: "The model simulations were run for the time period 1582-2008, a total of 427 years." – The boundary condition begins in 1948, so it can't be said that the model began in 1582. This was only an artifice used to simulate the same period for seven times.

15. L224: "the simulations $(CV+PC) - (CA+PC) = (CV-CA)|_{PC}$" → "the simulations $(CV+PC)$ and $(CA+PC)$" - The notation "$(CV-CA)|_{PC}$" is interesting, but it wasn't used. Then it can be removed.

16. L228: "and $CA+PC$, so that $(CA+PC+F)-(CA+PC) = F|_{CA,PC}$. Similarly," → "and $CA+PC$. Similarly,".

17. L230: "between $CV+PC+F$ and $CV+PC$, so that $(CV+PC+F)-(CV+PC) = F|_{CV,PC}$. The different" → "between $CV+PC+F$ and $CV+PC$. The different".

18. L273: "TB declined by 2% for PR", - In Figure 4b it looks positive, so it would be an increase instead of a decrease. Please check it.

19. L393: "compared to $CV+PC$" → "compared to $CV+PC - CA+PC$".

20. Figure 4d: "$CV + PG + F \ - \ CV + PC$" $\rightarrow$ "$CV + PG + F \ - \ CV + PG$".
* * *

---

## Referee Comment (RC3) · Anonymous Referee #3 · 25 Jul 2017

General Comment This manuscript mainly focuses on Amazon-Cerrado transitional vegetation. For this region, a mechanistic model is used to determine the effects of various processes on aboveground biomass (AGB). In particular, the effects of fire, phosphorus (P) limitation, and interannual climate variability are evaluated. It is concluded that all of these effects are important, but that fire is the main driver of vegetation change along the transition. The manuscript also reports that the model simulates >80% of the spatial variability in AGB in the transition zone.

Understanding the spatial distribution of tree biomass in the tropics is a very active area

of research. The questions asked by the authors, especially in regard to P limitation, are open ones and quite worthy of investigation. It is very reasonable to approach questions about mechanisms, such as those asked by the authors, using a mechanistic model. Nevertheless, I was unconvinced by the analysis that was presented. I have major concerns about the implementation of P limitation and the statistical analysis. The phenology scheme was not described in much detail, but could strongly influence the results. Several claims in the discussion were weakly, if at all, supported by the results.

Specific Comments 1. I do not think that the authors really implemented P-limitation in their model. As I understood the manuscript, simulated P dynamics do not affect vegetation biomass. Instead, the authors prescribe a map of Vmax based on a statistical regression between Vmax and soil P. As such, there is no mechanistic representation of P limitation in this model. Without a mechanistic link, I do not think it is correct to ascribe variation in AGB to variations in P. This manuscript can be improved by investigating the effects of different mechanistic implementations of P cycling and P limitation on the simulated vegetation.

2. Ptotal may indeed have some positive relation with Vmax, but Equation (1) still seems problematic to me. What happens when Ptotal is very large, and vegetation is presumably no longer limited by P? This equation would say that Vmax would still increase, but surely there must be some maximum value when other factors become limiting. More generally, I was not convinced that the most important way P affects plants is through Vmax. For example, what about maintenance of some approximate C:N:P stoichiometry, carbon costs of P acquisition, etc.?

3. The statistical analyses are inappropriate and do not support the conclusions. The statistical tests used by the authors are only appropriate when there is some random variable. I did not identify anything in the simulation design that could lead to a random effect (for example, some stochastic process). I recommend cutting the whole statistical analysis.

[Figure]

4. The model description incomplete. Is the source code available somewhere? Exactly how does this version of the model differ from previous versions? Any new equations or new parameter values need to be documented here.

5. The manuscript indicates (lines 120-121) that a temperature-based phenology scheme was used. But a drought phenology scheme is more appropriate for the tropics. I can imagine that the results would change dramatically if a drought phenology scheme were used. The original IBIS model had a drought phenology scheme, right? I guess that was not implemented here?

6. I was surprised that the manuscript did not discuss alternative stable states in terms of either the AGB database or the simulations. How was the AGB database constructed, given that there may be alternative stable states? I found it remarkable that the model was able to capture 80% of the variability. Would this result indicate that the idea of alternative stable states is not really appropriate in the Amazon-Cerrado transition?

Additional comments Lines 88-89: This is too vague. A discussion of the failures would be welcome.

Line 155: This equation needs more description. Is the same Vmax assigned to all PFTs? Is it meters square of leaf area or meters squared of ground?

Lines 269-276: It is arbitrary as to whether there are increases or decreases. Whether there is an increase or a decrease depends on the chosen baseline. Also on this paragraph, I am wondering whether tree biomass simply follows soil P?

Line 280: Note that the word "inflammable" actually means easily ignited.

Lines 278-281: Not justified. Where is water availability shown, and how is it defined?

Lines 300-302: Why does fire cause LAI to increase?

Line 409-411: There are exceptions (Goll et al, Yang et al).

Lines 416-424: This paragraph seems too speculative given the model results.

Lines 425-428: This is also not strongly supported.

Lines 460-462: But does it help explain the spatial variability?

Line 502: Showing the climate data would make this point more convincing.

---

## Author Comment (AC1) · 31 Jul 2017

(Reviewer comments in *italics;* Responses in **bold**)

Response to Anonymous Referee #1

*General Comments:*
*Reviewer summary:*
*The manuscript presents results from the DGVM INLAND evaluating the effect of climate variability, fire and phosphorus limitation on vegetation structure and dynamics in the transitional zone between the forested Amazon and grassland savanna of the Cerrado. Changes in net primary production, aboveground biomass, and leaf area index are assessed between simulations, and simulated aboveground biomass is compared to observations. Transects along the Amazon-Cerrado transition zone are analyzed in subsets, as well as the region as a whole. Inclusion of climate variability, fire and maps phosphorus limitation improves simulation of vegetation structure across the region and for four of five transects. The cerrado transect has the lowest correlation to observations. Fire has the strongest impact on vegetation characteristics, followed by phosphorous limitation. Overall, INLAND with these included factors appears effective at simulating vegetation across the region, but regional deficiencies show that more improvements can be made.*

*Article contribution and overall impact:*

*This study highlights the need for improved simulation of vegetation in a key forest-savanna transition zone. A shift in vegetation in this region has the capacity to impact the cycling of water and nutrients as well as energy fluxes beyond the area of forest-savanna transition. Uncertainty in future climate and fire behavior as well as the feedback between these factors make the vegetation state of this region difficult to predict. The manuscript does a good job of presenting the challenges of simulating vegetation in this transitional zone where climate, nutrients and fire are essential contributors to vegetation state. The inclusion of phosphorous limitation in the simulations for the region is an important addition to the evaluation of vegetation state. The discussion would benefit from a more detailed description of the fire model and fire activity during simulation. The concluding recommendations for future work also need to be clarified. Discussion of the importance of vegetation size structure and how this is or is not represented in INLAND should also be added. A key component of the mortality of woody vegetation to fire is its size at the time of fire and the ability to accumulate size between fires. This is central to the work of many of W. Hoffman's papers in the region (Hoffman et al 2003, Hoffman et al 2009, Hoffman et al 2012). Please add discussion of size structure to the manuscript.*

**Response: We are grateful to the reviewers for their insightful comments and helpful suggestions. We can include a better fire model description and describe how it works throughout the discussion of its impact on the vegetation structure. We will include a more complete discussion about the influence of fire on size structure along the transition. Details about recommendations for future work also may be clarified.**

*Detailed comments:*

1. *Page 6 line 101-107: Add the 1x1 degree grid size to this section.*

**Response: This information was added. See below:**
**"The present study focuses on the Amazon-Cerrado transition (Figure 1). We use the official delimitation of the Brazilian biomes proposed by IBGE (2004), and define five transects along the transition border with $1° \times 1°$ grid size (the terms "transition", "Amazon-Cerrado transition" and "Forest-Savanna transition" are used interchangeably with the same meaning throughout this manuscript)."**

2. *Page 9 line 159-164: Provide more detail concerning fire model. Is the fire ignition probability the same in every pixel, or is there spatial variability? If there is spatial variability what drives this parameterization? Does flammability vary by PFT?*

**Response: The fire ignition probability is spatially varied in INLAND. INLAND incorporates all fire components of the CTEM (Canadian Terrestrial Ecosystem Model) model (Arora and Boer, 2005). These components simulate fire at the daily timescale (instead of the yearly timescale of earlier models) by computing the probability of fire occurrence, which is based on biomass availability, flammability and ignition source for each pixel (using observed lighting frequency). Burned area is modeled as an ellipse of dimensions determined by wind and fuel conditions. The fire model of CTEM uses an arbitrary anthropogenic fire probability which is summed to the natural ignition probability. In summary, the natural ignition probability is represented by a lightning scalar, which varies from 0 to 1 as cloud-to-ground lightning frequency varies from a specified lower value of essentially no lightning to an upper value close to the maximum observed. The probability of fire ignition due to human causes may be selected depending on location and human activity and determines the lower limit of ignition constraint. In INLAND, we use 0.50 for the probability of fire ignition due to human cause. Additionally, we suggest the reviewer to verify the arbitrary ignition scheme in Figure 3c in Arora and Boer (2005), which we do not have permission to reproduce. Moreover, in INLAND the vegetation structure is represented by two layers. The fire occurrence reduces the upper canopy decreasing the trees LAI (LAI upper), and opening the canopy. The opening canopy implicates in more luminosity penetration and consequently increase of photosynthesis rates by grasses (increasing of LAI lower) initializing a competition between PFTs for light resource. We will include these details concerning fire model in the Methods section.**

3. *Page 16 line 316: define "transition" here and throughout manuscript. The reference is not always clear. Most often it appears to be Amazon-Cerrado transition or forest-savanna transition, if that is correct please add the extra detail here and throughout manuscript (Pg 20 line 403, pg 21 line 419, pg 22 line 426,428,443, pg 23 line 451)*

**Response: We agree with the reviewer that this term was not always clear and may generate doubts. To clarify, we included in Page 6 line 105-106 a brief reference: "The terms Amazon-Cerrado transition, forest savanna transition or transition are used interchangeably with the same meaning throughout this manuscript".**

4. *Page 17 line 323-325: Update sentence to "responsible for altering the simulated AGB to approach the observed AGB" or some variant. Current sentence structure is unclear.*

**Response: The sentence has been changed. See below:**
**"In T2, T3 and T4, however, fire is responsible for altering the simulated AGB to the observed AGB in the eastern pixels of the Cerrado domain (Figure 5)".**

5. *Page 18 line 345-349: What observational data set is this being compared to for current vegetation state? Is this a by pixel comparison of the same grid size?*

**Response: We use the official delimitation of the Brazilian biomes proposed by IBGE (2004), which reconstructs in more detail the probable location of the vegetation before the anthropogenic interference. This database is a vector database and is available in http://downloads.ibge.gov.br/downloads_geociencias.htm.**

6. *Page 18 line 359-364: Do the climate datasets used in simulation include reduced rainfall and ENSO effects? Explain this further.*

**Response: The CRU climate database, which is used in our study, included reduced rainfall and ENSO effects. The CRU gridded climate dataset is a database developed from monthly observations at meteorological stations across the world's land areas (Harris et al., 2014). These dataset includes six mostly independent climate variables (mean temperature, diurnal temperature range, precipitation, wet-day frequency, vapor pressure, and cloud cover). Maximum and minimum temperatures have been arithmetically derived from these. This gridded product will be publicly available, including the input station series (http://www.cru.uea.ac.uk/ and http://badc.nerc.ac.uk/data/cru/).**

7. *Page 21 line 409: Update to "for the most part Dynamic Vegetation Models"*

**Response: That has been changed.**

8. *Page 22 line 435-437: Add more detail clarifying how the INLAND model differs from reality. Is it able to simulate rapid restoration following fire? If not, what would need to be added to the model's fire or vegetation characteristics?*

**Response: In our model, the restoration of vegetation after fire occurrence is exclusively due to the canopy opening and consequently more luminosity penetration into lower canopy. Two layers in the INLAND represent the vegetation structure. The fire occurrence reduces de upper canopy decreasing the trees LAI (LAI upper), and opening the canopy. Thus, we have more luminosity penetration and consequently increase of photosynthesis rates by grasses (increase of LAI lower) initializing a**

competition for light resource. The dynamic vegetation module computes for each PFT: gross and net primary productivity, changes in biomass pools, simple mortality disturbance processes and resultant LAI, with the same manner before the fire occurrence, thus allowing vegetation type and cover to change with time.

The partitioning of the NPP for each PFT resolves carbon in three AGB pools: leaves, stems and fine roots. The LAI of each PFT is obtained by simply dividing leaf carbon by specific leaf area, which in INLAND is considered fixed (one value) for each PFT.

INLAND has eight soil layers to simulate the diurnal and seasonal variations of heat and moisture. Each layer is described in terms of soil temperature, volumetric water content and ice content (Foley et al., 1996; Thompson and Pollard, 1995). Furthermore, all of these processes are influenced by soil texture and amount of organic matter within the soil profile.

Considering these aspects of vegetation dynamics and soil physical properties the model can simulate plant competition for light and water between trees, shrubs and grasses through shading and differences in water uptake (Foley et al., 1996). We included more details about restoration following fire.

9. *Page 22 line 444: Update to "but the inclusion of these effects"*

**Response: That has been changed.**

10. *Page 22 line 442-445: This needs more explanation. Is the vegetation simulation insufficient because of the presence of transitional and robust pixels in the cerrado in fig 5? Or is this because of comparisons to observed data of vegetation in the cerrado?*

**Response: The sentence was re-written. See below:**
**"From Figure 5, it is clear that CV, F and P limitation in the transition zone reduce the AGB, approaching the simulated to the observed data. However, the inclusion of these effects is still insufficient to represent the actual vegetation structure in the Amazon-Cerrado border (Figure 6L)."**

11. *Page 23 line 449: what is meant by residence time?*

**Response: Residence time is the average time a particle resides (passes) in a pool or system. In INLAND, the residence time is a parameter of the vegetation used by the PFTs to allocate carbon in the different compartments of biomass (leaves, roots and stems). The residence time of carbon in a biomass compartment is intended to represent the loss of biomass through mortality and tissue turnover.**

12. *Page 24 line 477-479: Explain this in more detail: "It does not dynamically change the allocation"*

**Response: Parameters can be fixed or dynamically allocated, which means that they can change over time. Fixed parameters for a given PFT are assumed during the carbon allocation in INLAND. For tropical evergreens trees, for example, we have**

50% of carbon allocation in stems, 25% in leaves and 25% in roots. Even though there is evidence that in the Amazonia-Cerrado transition the carbon allocation rates may vary in some situations of water stress, the INLAND model do not represent this strategy.

*13. Page 25 line 495-496: Reword this sentence. The meaning is not clear.*

**Response: We agree to the reviewer. This sentence was changed. See below: "Obtaining ecophysiological parameters is a challenge to the scientific community once the field measurements depend on fieldwork conducted throughout all the transition area."**

*14. Page 26 line 522-525: Inclusion of spatially explicit parameters may or may not improve DGVM simulation. This assumes that the existing processes are accurate, and that it is merely parameters. Provide more discussion of this possibility, or reword this section.*

**Response: There are evidences that the inclusion of spatially explicit parameters may improve DGVM simulation. Castanho et al. (2013) showed that the simulated aboveground biomass in Amazonia improved with spatially biophysical parameters such as woody biomass residence clearer time, maximum, carboxylation capacity (Vmax), and NPP allocation to wood. They found that using single values for key ecological parameters in the tropical forest biome severely limits simulation accuracy. We believe that the same limitation occurs along the transition. The use of spatial parameters allows represent the spatial heterogeneity along the Amazon-Cerrado border and may lead to simulated spatial variability of biomass. However, the lack of data observed in this region limits the representation and understanding of how biophysical parameters vary throughout the transition. Thus, it is necessary obtain physiological and structural parameters to establish numerical relationships between the environment and the vegetation dynamic models.**

*Page 26 line 525-527: What is meant by temporal variability? Size structure?*

**Response: Currently the physiological parameters of vegetation are fixed, each PFT uses a fixed carbon allocation parameter, mortality, carboxylation capacity and others from start to finish of the simulation. We suggest that these parameters should be dynamically allocated, i.e. temporal variability in physiological parameters of vegetation, as a function of other simulated variables should be included, to improve the simulation results.**

---

## Author Comment (AC2) · 31 Jul 2017

Reviewer comments in *italics;* Responses in **bold**)

Response to Anonymous Referee #3

General Comment: *This manuscript mainly focuses on Amazon-Cerrado transitional vegetation. For this region, a mechanistic model is used to determine the effects of various processes on aboveground biomass (AGB). In particular, the effects of fire phosphorus (P) limitation, and interannual climate variability are evaluated. It is concluded that all of these effects are important, but that fire is the main driver of vegetation change along the transition. The manuscript also reports that the model simulates >80% of the spatial variability in AGB in the transition zone.*
*Understanding the spatial distribution of tree biomass in the tropics is a very active area of research. The questions asked by the authors, especially in regard to P limitation, are open ones and quite worthy of investigation. It is very reasonable to approach questions about mechanisms, such as those asked by the authors, using a mechanistic model. Nevertheless, I was unconvinced by the analysis that was presented. I have major concerns about the implementation of P limitation and the statistical analysis.*
*The phenology scheme was not described in much detail, but could strongly influence the results. Several claims in the discussion were weakly, if at all, supported by the results.*

*Specific Comments*

*I do not think that the authors really implemented P-limitation in their model. As I understood the manuscript, simulated P dynamics do not affect vegetation biomass. Instead, the authors prescribe a map of Vmax based on a statistical regression between Vmax and soil P. As such, there is no mechanistic representation of P limitation in this model. Without a mechanistic link, I do not think it is correct to ascribe variation in AGB to variations in P. This manuscript can be improved by investigating the effects of different mechanistic implementations of P cycling and P limitation on the simulated vegetation.*

**Indeed, we did not implement a full P cycle model, but rather we parametrized what we believe is the most important relationship between soil P and photosynthesis. In this study, we only use the P-limitation through the linear relation developed by Castanho et al. (2013) to evaluate the biomass along the Amazon-Cerrado transition. The P cycle is slow, and a full representation of the soil P may be advantageous only over a time scale of several decades to centuries. In the shorter term (a few decades), starting from a known point in soil fertility as we did, seems to be a more appropriate approach. The values of P in the map influence Vmax, which mechanistically influences NPP and biomass. This has been demonstrated before by Castanho et al. (2013).**
**The implementation of a full P cycling and is out of the scope of this manuscript. The main objective of this manuscript is investigate the influence of P-limitation, climate and fire occurrence on AGB, and not implement P cycling.**

1. *Ptotal may indeed have some positive relation with Vmax, but Equation (1) still seems problematic to me. What happens when Ptotal is very large, and vegetation is presumably no longer limited by P? This equation would say that Vmax would still increase, but surely there must be some maximum value when other factors become limiting.*

Equation (1), being an empirical equation, represents the relationship between P and $V_{cmax}$ in the range where there are observed values. In the hypothetical case of a very high Ptotal, and a consequent very high Vmax, other bottlenecks in the model code (limitation by light or by water or by temperature) would limit gross photosynthesis. However, this is unlikely, given that the Amazon and Amazon-Cerrado transition soils are known for being nutrient poor.

*More generally, I was not convinced that the most important way P affects plants is through Vmax. For example,* what about maintenance of some approximate C:N:P stoichiometry, carbon costs of P acquisition, etc.?

**Castanho et al. (2013) provided empirical evidence that Vmax is influenced by P-total. The other processes described by the reviewer, along with other process not described, may also be important, but they have yet to be implemented and tested. Our study, however, falls in line with the common use of parameterizations in the modeling literature, to quickly test some effects before a full and more complex model is implemented.**

2. *The statistical analyses are inappropriate and do not support the conclusions. The statistical tests used by the authors are only appropriate when there is some random variable. I did not identify anything in the simulation design that could lead to a random effect (for example, some stochastic process). I recommend cutting the whole statistical analysis.*

**Response: With all due respect to the reviewer, we believe that this point is not correct. There are several sources of random variability in the data, such as (from lower to higher): spatial variability in the soils texture (in both CA and CV simulations), spatial variability in soil fertility (in PR and PG simulations), the interannual climate variability (in the CV case), and the random ignition in the fire module.**

3. *The model description incomplete. Is the source code available somewhere? Exactly how does this version of the model differ from previous versions? Any new equations or new parameter values need to be documented here.*

**Response: The source code can be downloaded from http://biosfera.dea.ufv.br/en-US, clicking on models and, then, on INLAND. The INLAND project was mainly a revision of the IBIS code, through assembly and standardization of different IBIS versions, and improvements in software engineering. We used the version described by Senna et al. (2009) as starting point for INLAND. No changes in tuning were done since that paper, except the addition of the P parameterization, which was described in detail. Some of the key equations and parameterizations, however, were described by Foley et al. (1996) and Kucharik et al. (2000).**

4. *The manuscript indicates (lines 120-121) that a temperature-based phenology scheme was used. But a drought phenology scheme is more appropriate for the tropics. I can imagine that the results would change dramatically if a drought phenology scheme were used. The original IBIS model had a drought phenology scheme, right? I guess that was not implemented here?*

**Response: We agree and thanks the reviewer for this commentary. That was our mistake. This sentence has been modified. The INLAND, such as IBIS, has phenology scheme based on winter-deciduous and drought-deciduous behavior of particular PFTs. For winter-deciduous plants (outside the area of study), the temperature based is used, while for tropical drought-deciduous plants, the PFTs drop their leaves during the least productive months of the year, defined in terms of the previous year's carbon balance. This phenology scheme is extensively described in Foley et al. (1996) and Kucharik et al. (2000). Lines 120-121 were changed to "The vegetation phenology module simulates the processes such as budding and senescence based on drought phenology scheme for tropical deciduous trees."**

5. *I was surprised that the manuscript did not discuss alternative stable states in terms of either the AGB database or the simulations. How was the AGB database constructed, given that there may be alternative stable states? I found it remarkable that the model was able to capture 80% of the variability. Would this result indicate that the idea of alternative stable states is not really appropriate in the Amazon-Cerrado transition?*

**Response: We are not sure we understand this comment. Alternatives states in this region would arise as a consequence of climate change and antropogenic disturbance. This is clearly outside the scope of this study, which aims at representing the actual non-disturbed status of the vegetation, and attribute the processes responsible for its present status, in terms of climate variability, fire and P limitation.**

*Additional comments*

*Lines 88-89: This is too vague. A discussion of the failures would be welcome.*

**Response: In accordance with the referees's wishes, we included references and have now changed this paragraph to "Currently, no model has demonstrated to be able to accurately simulate the vegetation transition between Amazon and Cerrado. In general the DGVMs simulate evergreen forest along the Amazon-Cerrado border neglect savanna occurrence (Botta and Foley, 2002; Bond et al., 2005; Salazar et al., 2007; Smith et al., 2014). This difficulty may be due to absence or not well represented disturbances such as fire, nutritional limitation or soil proprieties. Thus, we need a better understanding of the drivers on transitional vegetation to determine the parameters and establish relations between the environmental and transitional vegetation physiognomies."**

*Line 155: This equation needs more description. Is the same Vmax assigned to all PFTs? Is it meters square of leaf area or meters squared of ground?*

**Response: There is reference to equation and specified the PFTs limited by phosphorus on the manuscript. Please check Line 157: "This equation has been developed by Castanho et al. (2013) based on data for tropical evergreen and deciduous trees, and is applied only to these two PFTs in the model."**

*Lines 269-276: It is arbitrary as to whether there are increases or decreases. Whether there is an increase or a decrease depends on the chosen baseline. Also on this paragraph, I am wondering whether tree biomass simply follows soil P?*

**Response: The increase or decrease biomass in PG and PR was evaluated from PC using the subtraction between the simulations: (CV+PR)−(CV+PC) and (CV+PG)–(CV+PC). These differences represents the isolated effect of P limitations on tree biomass along the simulated area.**

*Line 280: Note that the word "inflammable" actually means easily ignited.*

**Response: We agree with the reviewer. This sentence was changed to "The small or null fire effect in the Central Amazon rainforest is related the greater water availability on the Amazonia, which makes the forest naturally not flammable as well as a gradient towards seasonally dryer climate increases the intensity and magnitude of fire effects towards the Cerrado (Figure 4d)."**

*Lines 278-281: Not justified. Where is water availability shown, and how is it defined?*

**Response: We agree with the reviewer. This sentence was changed to "The small or null fire effect in the Central Amazon rainforest is related the greater water availability on the Amazonia, which makes the forest naturally not flammable as well as a gradient towards seasonally dryer climate increases the intensity and magnitude of fire effects towards the Cerrado (Figure 4d)."**

*Lines 300-302: Why does fire cause LAI to increase?*

**Response: In INLAND the vegetation structure is represented by two layers: upper and lower canopy. The fire occurrence reduces the upper canopy decreasing the trees LAI (LAI upper), opening the canopy, and leading to more luminosity available to the lower layer, and consequently increasing the photosynthesis rates by grasses (increase of LAI lower). In general, the LAI lower increases quickly after the fire occurrence contributing with the increases of the LAI total that represents the sum of the two layers.**

*Line 409-411: There are exceptions (Goll et al, Yang et al).*

**Response: We agree to the reviewer. This sentence was changed.**

*Lines 416-424: This paragraph seems too speculative given the model results.*

**Response: We excluded this paragraph.**

*Lines 425-428: This is also not strongly supported.*

**Response: This is supported by Tables 3, 4 and 5.**

*Lines 460-462: But does it help explain the spatial variability?*

**Response: Yes. Patterns along the eastern part of the transects are determined by fire, which is less important in the western part of the transects.**

*Line 502: Showing the climate data would make this point more convincing.*

**Response: This is well known in the literature, but anyway. We provide three figures in this response to clarify this point to the reviewer. The seasonality of precipitation for Amazon and Cerrado biomes used in this study is shown in Figure R1. The dry season duration is larger in the Cerrado domain (Figure R1a) than in the Amazonia domain (Figure R1b). In the Cerrado, dry season comprise a period of about 6 months with little or no rain.**

**Spatial variability of precipitation and temperature are shown in Figures R2 and R3, respectively. These figures plot the difference between the average of 1999-2008 (a subset of CV) and CA (average of 1961-1990) highlighting the spatial variability of these climate variable throughout the study area. Comparing the interannual climate variability with the average climate, precipitation decreases (Figure R2) and temperature increases up 1.5°C (Figure R3) in central Cerrado in October, November, December and January. The lower precipitation associated with higher temperatures in central Cerrado can explain a low biomass, low LAI vegetation and savanna existence without fire disturbance. Note that this is a 10-year subset of the CV database. The actual year-to-year variations present much more intense amplitudes.**

[Figure]

**Figure R1. Seasonal of precipitation for Amazon and Cerrado domains for average climate (CA) - black line - and the last ten years of interannual- climate variability (CV) – color lines.**

[Figure]

**Figure R2. Spatial variability of precipitation for study area considering the average of the last 10 years of CV (1999-2008) and average climate CA (1961-1990).**

Spatial variability of temperature (CV - CA)

[Figure]

**Figure R3. Spatial variability of temperatures for the study area considering the average of the last 10 years of CV (1999-2008) and average climate CA (1961-1990).**

**References:**

Bond, W. J., Woodward, F. I. and Midgley, G. F.: The global distribution of ecosystems in a world without fire, New Phytol., 165(2), 525–538, doi:10.1111/j.1469-8137.2004.01252.x, 2005

Smith, B., W\a arlind, D., Arneth, A., Hickler, T., Leadley, P., Siltberg, J. and Zaehle, S.: Implications of incorporating N cycling and N limitations on primary production in

an individual-based dynamic vegetation model, Biogeosciences, 11(7), 2027–2054, doi:10.5194/bg-11-2027-2014, 2014

---

## Author Comment (AC3) · 2 Aug 2017

(Reviewer comments in *italics;* Responses in **bold**)

Response to Anonymous Referee #2

*This work uses the vegetation model INLAND to evaluate the individual and combined effects of the climate variability, the fire and the Phosphorus (P) limitation on the Brazilian ecosystem. The changes on the NPP, ] and AGB were evaluated in relation to 12 climate simulations. The AGB was also evaluated in function of observed data. In addition to climate variability, this work shows the importance of considering the soil nutrient limitation as well as the disturbances caused by the biomass burning in the study of vegetation dynamics. It is also presented some deficiencies of the DGVMs and the databases used to feed the INLAND model. Understanding the mechanisms that affect the vegetation and the efforts to improve numerical models in order to simulate such effects is of paramount important to the scientific progress. Therefore, this study is of great relevance and, in my opinion, it is suitable for publication in the*
*BG. However, I have some recommendations and doubts that I would like to see being clarified before publication.*

***Specific comments***

1. *L55-L60: observing the Figure 6d (CV+PC), all the Amazon region became "very robust", so we can assume that the simulation that considers only the climatic effect didn't indicate the "savannization of the Amazon", in other words, the results obtained in this study don't agree with the mentioned works. Can you comment on this?*

**Response: The "savannization of the Amazon" have been appointed by studies that use future climate scenarios (Oyama and Nobre, 2003; Betts et al., 2004; Cox et al., 2000; Cox et al., 2004; Salazar et al., 2007 Pereira et al., 2012) or climate scenarios associated to changes on vegetation in the deforestation case (Shukla et al., 1990; Malhi et al., 2009; Pires and Costa, 2013). Our results do not indicate the "savannization of the Amazon" because they are based on past climate.**

2. *L140: "values smaller than 0.8 $m^2m^{-2}$ characterize a grassland vegetation type" – Grassland can have LAI values much higher than 0.8 $m^2m^{-2}$. Darvishzadeh et al., 2008 found out grassland's average values of 2.76 $m^2m^{-2}$ and maximum value of 7.34 $m^2m^{-2}$. Please check if the INLAND really utilizes this threshold of LAI to define grassland.*

**Response: We agree with the reviewer, although we think the reviewer misunderstood the text. The total LAI for grasslands can definitely be much higher than 0.8 $m^2m^{-2}$. In this sentence, we refer to the LAI upper (tree LAI). That means that grassland vegetation type cannot have trees with LAI greater than 0.8 $m^2m^{-2}$ while the LAI lower (grasses) is not limited.**

3. *3. L85-L87: According to Oliveira et al. (in press), the weather also has influence in the nutrients. Then, the climate change's effect cannot be higher due to the indirect effects in the nutrients? Can you comment on this?*

**Response: Certainly, there is influence between weather and nutrients. For example, a very intense rain can leach nutrients, such as nitrogen, as well as strong winds can carry clay particles where numerous nutrients are adsorbed. However, in this work the nutritional conditions are prescribed and fixed. In this way, we cannot assert that the climate change's effect is higher due to the indirect effects in the nutrients. Oliveira et al (in press) has also been updated to Oliveira et al. (2017).**

4. *L157-L158: How are the other PFTs affected by the availability of P?*

**Response: Tropical evergreen and deciduous trees are affected by availability of P in INLAND while other PFTs are unaffected. The lack of scientific reports about the influence of phosphorus on shrub and grass vegetation along the Amazon basin justify why only these two PFTs are affected. The phosphorus limitation in INLAND is based on regression equation developed by Castanho et al. (2013), which use results of Fyllas et al. (2009) and Mercado et al. (2009). Fyllas et al. (2009) show that soil fertility is one of the most important predictors for observed higher nutrient concentration in Amazon tree leaves, while Mercado et al. (2009) report that the correlation between observed $V_{max}$ and P concentration in Amazon tree leaves. Thus, Castanho et al. (2013) developed a similar regression equation to that of Mercado et al. (2009, 2011) between $V_{max}$ and total P concentration in soil, instead of P concentration in leaves, allowing estimate $V_{max}$ for the whole Amazon.**

5. *L262: I didn't understand where the 8.7% came from. Could you make it clearer?*

**Response: We calculated the average biomass values for each biome area (Amazon and Cerrado) using the biomes delimitation from IBGE showed in Figure 1. This average biomass value was calculated for all simulations (Table 1), to compare the difference in the average value of each simulation. In Cerrado, average biomass is 8.7% lower in the simulation with inter-annual climate variability (CV) than with average climate (CA).**

6. *L339-L344: It can be seen in Figure 6 large differences between CA + F (Line3) and CV + F (Line 4). However, the differences between CA (Line 1) and CA + F (Line 3) and the ones between PC (Column 1), PR (Column 2) and PG (Column 3) are not very significant. Thus, the climatic variability is dominant when considering the three effects. Probably, if a fifth map showing CA + PG +F − CA+PG is constructed in Figure 4, it will be quite distinct from Figure 4d. Therefore, it should be exposed more clearly how it came up to the conclusion described in L513-515.*

**Response: In Figure 6, we evaluate only the robustness of the simulations. Although there is little difference between the vegetation types along the central Cerrado domain in CA (Line 1) and CA + F (Line 3), the most part of these changes is very robust when fire is considered. We do not use the vegetation types to infer which are the main determinant drivers on the transition, we use the Figure 6 only to show how these drives affect the distribution of vegetation. It is clear, however, that the CV+F**

combinations yield the best vegetation patterns, with minor differences associated with soil P.

The difference between CA + PG +F − CA+PG shows similar behavior that on CV simulations, but we did not plot this difference because in our statistical analyses only the interannual climate variability showed significant influence on AGB and LAI. Finally, to write the sentence in L513-515 ("fire is in the main determinant factor of the vegetation changes along the transition. The nutrient limitation is second in magnitude, stronger than the effect of inter-annual climate variability"), we use F-statistics in Tables 3, 4 and 5, which permits infer the magnitude of fire, phosphorus limitation, and climate on AGB and LAI. To clarify this, we rewrote this sentence to "Based on the F-statistic in Tables 3, 4 and 5, this work shows that fire is in the main determinant factor of the changes in vegetation structure (LAI, AGB) changes along the transition. The nutrient limitation is second in magnitude, stronger than the effect of inter-annual climate variability."

7. *L342: I think it is unlikely that an area with "deciduous forest" will turn into "evergreen forest" after being consumed by fire. Please comment if this is possible or if it is a model deficiency.*

Response: In INLAND simulation, the "deciduous forest" is turning into "evergreen forest" after being consumed by fire happens in only 5 pixels and in average climate condition only, in a clay soil with large water retention capacity (Figure 6G-I) In this situation, where there is little water stress in the CA simulation, both evergreen and drought deciduous PFTS have each one very high LAI, and the PFT that dominates can be defined by minor effects. Fire, although active, is probably too small to be relevant in a non-stressed ecosystem. However, when the interannual climate variability is considered, INLAND replaces the "evergreen forest" in Figures 6G-I to "Savanna and Grasslands" (Figure 6J-L). These results show the limitations of CA and the importance to consider the interannual climate variability on simulations to improve the vegetation simulated.

*Technical corrections*

1. *L22: "1960 – 1990" → "1961 – 1990", as described in L204.*
   **Response: This has been changed in the revised manuscript.**

2. *L23: "two regional datasets" → "two datasets".*
   **Response: This has been changed in the revised manuscript.**

3. *L62: "particularly the P limitation." → "particularly the Phosphorus (P) limitation."*
   **Response: This has been changed in the revised manuscript.**

4. L72: "Phosphorus (P) is a" → "P is a" or "Phosphorus is a".
   **Response: This has been changed in the revised manuscript.**

5. *L103: Transects 1 and 2 are more related to "Cerrado" than "Amazon", as shown in Figures 1, 2 and 5. Please rewrite this sentence.*
   **Response: We do not agree with this technical correction. The transects were located 50% in Amazon domain and 50% in Cerrado domain, as we described.**

6. L105: "Transect 1 (T1, 43 °-49 °W; 5 °-7 °S)" → "Transect 1 (T1, 44 °-50 °W; 5°-7°S)".
   **Response: This has been changed in the revised manuscript.**

7. L107: "Transect 5 (T5, 53 °-61 ° W; 13 °-15 ° S)" → "Transect 5 (T5, 52 °-60°W;13°-15 ° S)".
   **Response: This has been changed in the revised manuscript.**

8. L138: "annual mean $LAI_{upper}$ above" → "annual mean $LAI_{upper}$ below".
   **Response: We do not agree with this technical correction. The annual average of "LAI upper" in INLAND needs to be above 0.8 $m^2$ $m^{-2}$. This value is prescribed by the model.**

9. L173: "We used the P-mehlich-1" → "We used the Pmehlich-1".
   **Response: This has been changed in the revised manuscript.**

10. L176: "resulting in 12 additional pixels" – Wouldn't it be 6?
    **Response: There are 12 pixels, as described in Suplementary Materials.**

11. L176: "pixels with observed total P content" → "pixels without observed total P content"
    **Response: These additional pixels with the observed total P content are described in Suplementary Materials.**

12. L186: "Above-Ground AGB (AGB) database" → "Above-Ground Biomass (AGB database"
    **Response: This has been changed in the revised manuscript.**

13. L194-L196: There are two pixels for each longitude in each transects. Do the Figures 3 and 5 show the mean of the two pixels, or only the upper or the lower one?
    **Response: The Figures 3 and 5 show the average of two pixels. This information was added to the caption of both figures.**

14. L212: Remove the phrase: "The model simulations were run for the time period 1582-2008, a total of 427 years." – The boundary condition begins in 1948, so it can't be said that the model began in 1582. This was only an artifice used to simulate the same period for seven times.
    **Response: This has been changed in the revised manuscript.**

15. L224: "the simulations $(CV +PC) − (CA+PC) = (CV −CA)|PC$" → "the simulations $(CV +PC)$ and $(CA+PC)$" - The notation "$(CV −CA)|PC$" is interesting, but it wasn't used. Then it can be removed.
    **Response: This has been changed in the revised manuscript.**

16. L228: "and $CA+PC$, so that $(CA+PC+F)−(CA+PC) = F|CA,PC$. Similarly,"→ "and $CA + PC$. Similarly,".
    **Response: This has been changed in the revised manuscript.**

17. L230: "between $CV +PC+F$ and $CV +PC$, so that $(CV +PC+F)−(CV +PC) =F|CV,PC$. The different" → "between $CV +PC +F$ and $CV +PC$. The different".
    **Response: This has been changed in the revised manuscript.**

18. L273: "TB declined by 2% for PR", - In Figure 4b it looks positive, so it would be an increase instead of a decrease. Please check it.
    **Response: This 2% refers to the area average of Cerrado domain.**

19. L393: "compared to $CV + PC$" → "compared to $CV + PC − CA + PC$".
    **Response: In this sentence, we are comparing CV+PG – CV+PC.**

20. Figure 4d: "$CV + PG + F − CV + PC$" → "$CV + PG + F − CV + PG$".
    **Response: This has been changed in the revised manuscript. Although this is not a regionally relevant event, we modified the text to include this discussion.**

 **References:**

**Fyllas, N. M., Pati˜no, S., Baker, T. R., Bielefeld Nardoto, G., Martinelli,L. A., Quesada, C. A., Paiva, R., Schwarz, M., Horna, V., Mercado, L. M., Santos, A., Arroyo, L., Jim´enez, E. M., Luiz˜ao,F. J., Neill, D. A., Silva, N., Prieto, A., Rudas, A., Silviera, M.,Vieira, I. C. G., Lopez-Gonzalez, G., Malhi, Y., Phillips, O. L.,and Lloyd, J.: Basin-wide variations in foliar properties of Amazonian forest: phylogeny, soils and climate, Biogeosciences, 6,2677–2708, doi:10.5194/bg-6-2677-2009, 2009.**

**Mercado, L. M., Lloyd, J., Dolman, A. J., Sitch, S., and Pati˜no,S.: Modelling basin-wide variations in Amazon forest productivity– Part 1: Model calibration, evaluation and upscaling functionsfor canopy photosynthesis, Biogeosciences, 6, 1247–1272, doi:10.5194/bg-6-1247-2009, 2009.**

Mercado, L. M., Pati~no, S., Domingues, T. F., Fyllas, N. M., Weedon,G. P., Sitch, S., Quesada, C. A., Phillips, O. L., Arag~ao, L.E. O. C., Malhi, Y., Dolman, A. J., Restrepo-Coupe, N., Saleska,S. R., Baker, T. R., Almeida, S., Higuchi, N., and Lloyd, J.: Variations in Amazon forest productivity correlated with foliar nutrients and modelled rates of photosynthetic carbon supply, Philos. T. Roy. Soc. B, 366, 3316–3329, doi:10.1098/rstb.2011.0045, 2011.

---

## Author Response (AR2)

(Reviewer comments in *italics;* Responses in **bold**)

**Response to Report#1** Received: 10 Oct 2017

*Suggestions for revision or reasons for rejection*

*General Comments:*
*Revision: The revision has addressed many concerns and comments, but there are still changes to be made prior to publication.*

*Detailed comments:*

*1-     Add reference for Goll et al 2012*
**Response: Included, please check line 725 in marked-up manuscript version.**

*2-     Page 30 line 781-783: Provide a better description of how pools are effected by fire. Arora and Boer do not use the term "penalization fraction", and this term is not clear.*
**Response: We clarify this in the revised manuscript, please check section 2.2.**

*3-     Page 38 line 985-986: Update sentence to "responsible for altering the simulated AGB to approach the observed AGB". Current sentence structure is unclear.*
**Response: This has been changed in the revised manuscript. Please check lines 172-180 in marked-up manuscript version.**

*4-     Page 42 line 1092-1094: Update sentence to "showed significant spatial differences".*
**Response: This has been changed in the revised manuscript. Please check line 461 in marked-up manuscript version.**

*5-     Page 43 line 1109-1112: Update to "vegetation-fire dynamics are mainly controlled"*
**Response: This sentence has been removed as suggested in Comment 6.**

*6-     Page 43 line 1109-1115: This section needs to be re-written to clearly present how fire effects AGB. Remove the comparison to the work of Hoffman. Arora and Boer clearly describe fire-related mortality with specific reference to tropical burning. Summarize this here. Specifically, does INLAND use the same combustion factors for stem, leaf, and root biomass discussed in Arora and Boer? (To quote Arora and Boer: "The stem wood combustion factor for tropical drought deciduous trees at 0.10 is smaller than that for other PFTs since these trees are characterized by bark that is 3 times thicker than other trees [Hoffmann et al., 2003] which makes them well-adapted for fire-prone savanna regions.")*

**Response: This has been rewritten in the revised manuscript. The fire occurrence is a disturbance applied equally for all PFTs inside the same pixel. Please check lines 478-488 in marked-up manuscript version.**

*7-      Page 44 line 1179-1183: This should be connected to the above paragraph, or include more discussion. Update to "fire effect implies significant increases" and "resulting in an increase of LAI$_{lower}$"*
**Response: This has been rewritten in the revised manuscript. Please check lines 485-488 in marked-up manuscript version.**

*8-      Page 44 line 1184: Remove double negative. Update to "model does not include characteristics related to"*
**Response: This has been changed in the revised manuscript. Please check lines 489-490 in marked-up manuscript version.**

*9-      Page 44 line 1186: Remove double negative. Update to "or the representation of some"*
**Response: This has been changed in the revised manuscript. Please check line 490 in marked-up manuscript version.**

*10-     Page 45 line 1243: Tree structure would vary temporally rather than spatially. It should be highlighted as an area for improvement, but not in a context of spatial variability.*
**Response: We agree with the reviewer, this sentence has been modified. Please check line 549 in marked-up manuscript version.**

*11-     Page 48 line 1362: Update to "There is evidence that…"*
**Response: This has been changed in the revised manuscript. Please check line 635 in marked-up manuscript version.**

*12-     Page 49 line 1370-1371: Update to "With the help of these data, dynamic vegetation models will be able to improve simulation of current…"*
**Response: This has been changed in the revised manuscript. Please check lines 643-644 in marked-up manuscript version.**

 Received: 10 Oct 2017

*All major content-related questions and recommendations have been followed. However, I think that some answers need to be incorporate into the manuscript to improve it. They are:*

*1- R1Q05 (Referee-1: Question-05)*
**Response: This was already explained in the first two sentences of Section 2.1.**

*2- R1Q06 (Referee-1: Questions-06)*
**Response: This has been included in the revised manuscript. Please check lines 240-247 in marked-up manuscript version.**

*3- R1Q12 (Referee-1: Question-12)*
**Response: This has been included in the revised manuscript. Please check lines 577-580 in marked-up manuscript version.**

*4- R2Q03 (Referee-2: Questions-03)*
**Response: The discussion was included in Section 4. Please check lines 468-471 in marked-up manuscript version.**

*5- R2Q04 (Referee-2: Questions-04)*
**Response: The other PFTs are unaffected. It was already explicit, please check lines 170-171 in marked-up manuscript version.**

*6- R3Q03 (Referee-3: Questions-03)*
**Response: This has been included in the revised manuscript. Please check lines 119-120 in marked-up manuscript version.**

*The answer to the last question from referee #3 should go to the supplementary material.*
**Response: This has been included in the supplementary material, Section S3.**

[revised manuscript text omitted]

**Comentado [E1]:** *1-This section needs to be re-written to clearly present how fire effects AGB. Remove the comparison to the work of Hoffman.* **Arora and Boer clearly describe fire-related mortality with specific reference to tropical burning. Summarize this here.** *Specifically, does INLAND use the same combustion factors for stem, leaf, and root biomass discussed in Arora and Boer? (To quote Arora and Boer: "The stem wood combustion factor for tropical drought deciduous trees at 0.10 is smaller than that for other PFTs since these trees are characterized by bark that is 3 times thicker than other trees [Hoffmann et al., 2003] which makes them well-adapted for fire-prone savanna regions.")*

**Comentado [E2]:** *2-This section needs to be re-written to clearly present how fire effects AGB. Remove the comparison to* …

**Comentado [E3]:**

**Comentado [E4]:** This should be connected to the above …

[revised manuscript text omitted]

---

## Author Response (AR3)

(Reviewer comments in *italics;* Responses in **bold**)

**Response to Associate Editor** Received: 20 Dec 2017

**We appreciate the suggestions. The manuscript has been completely revised by a professional editor to provide better wording for unclear sentences and bring the manuscript more in line with the style conventions in *Biogeosciences*. This includes a minor change in the title of the manuscript. Also, we reviewed the discussion and removed some non-essential points. However, most of the discussion that was explicitly requested by previous reviewers was retained. In addition, all referee´s suggestions have been included and are detailed below.**

**With these modifications, we hope you find that the manuscript is now acceptable for publication.**

*Line 22: Update to: "to an observed AGB map."*
**Response: This has been changed in the revised manuscript. Please check line 31 in marked-up manuscript version.**

*Lines 71-73: Revise "unlike the temperate forests" and "adsorvided"*
**Response: This has been changed in the revised manuscript. Please check lines 155-156 in marked-up manuscript version.**

*Lines 80-81: Revise: "However, N is not a limiting nutrient for trees in the tropics (Davidson et al. 2004), while P availability affects the trees dynamics."*
**Response: This has been changed in the revised manuscript. Please check line 163-165 in marked-up manuscript version.**

*How much does this AGB map differ from other AGB maps (e.g. from Mitchard et al., Saatchi et al.) and why did you choose to use this one? It might be helpful to add 1-2 sentences in the discussion about the uncertainties in the "observational" maps when comparing them to model results.*
**Response: Although Saatchi et al., (2009), Baccini et al., (2012) and Mitchard et al., (2014) are maps of pantropical biomass, they were for present day biomass, including deforested pixels. Deforestation is widespread in the region, compromising the comparison with our results, since the goal of this study is to understand the drivers of natural vegetation structure and dynamics in the transition zone. The same justification is valid for the Avitabil data, since this map represents the data combination of Saatchi and Baccini. Thus, we prefer to use the biomass database of Nogueira et al. (2015) that includes aboveground measures in the original vegetation (before extensive clearing), and uses 74 different classes of vegetation and better representing the many physiognomies in the region. The areas that are currently degraded were identified, and biomass was assigned according to information about the original vegetation.**

**Response to Report#1** Received: 20 Dec 2017

*General Comments:*
*Revision: The revision comments here are mainly associated with language and wording.*

*Detailed comments:*
*Abstract: line 25-26: Update phrase "gradually improve simulated vegetation types" What is meant by "improve" vegetation?*
**Response: This has been changed in the revised manuscript. Please check line 35 in marked-up manuscript version.**

*Page 3 line 40-42: Update "physiognomies" and "woodland formations" to something more clearly descriptive of vegetation.*
**Response: This has been changed in the revised manuscript. Please check lines 86-87 in marked-up manuscript version.**

*Page 3 line 42-44: Update to "ecotonal vegetation around this transition includes a mix of tropical forest and savanna species."*
**Response: This has been changed in the revised manuscript. Please check lines 88-89 in marked-up manuscript version.**

*Page 6 line 115: Update to "INLAND is a revision of the IBIS model"*
**Response: This has been changed in the revised manuscript. Please check line 268 in marked-up manuscript version.**

*Page 7 line 117-119: Update to "We used the version described by Senna et al. (2009) as the starting point for INLAND without changes in tuning, aside from the addition of P parameterization, described below."*
**Response: This has been changed in the revised manuscript. Please check lines 271-272 in marked-up manuscript version.**

*Page 7 line 131: Update to "based on a drought phenology scheme"*
**Response: This has been changed in the revised manuscript. Please check line 284 in marked-up manuscript version.**

*Page 8 line 152-153: Include a basic description of Cerrado and Camp sujo. Some papers describe cerrado as "savanna woodland with 10-60% tree cover".*
**Response: We used the Ribeiro and Walter, 2008 definition for both physiognomies. The reference to the definition was explicitly included in Line 374 of the marked-up manuscript version.**

*Page 15 line 293: Update to "is much higher in magnitude than due to P"*

**Response: This has been changed in the revised manuscript. Please check line 735 in marked-up manuscript version.**

*Page 15 line 294-297: Split this into two sentences.*
**Response: This has been changed in the revised manuscript. Please check lines 736-739 in marked-up manuscript version.**

*Page 18 line 344: Update to "cover in the last 10 years of simulation"*
**Response: This has been changed in the revised manuscript. Please check line 881 in marked-up manuscript version.**

*Page 18 line 350: Update to "deciduous forest with evergreen forest"*
**Response: This has been changed in the revised manuscript. Please check line 888 in marked-up manuscript version.**

*Page 18 line 360: Update to "PFTs each have a very high LAI"*
**Response: This has been changed in the revised manuscript. Please check line 898 in marked-up manuscript version.**

*Page 18 line 361: Explain what is meant by "minor effects" determining dominance, or delete this phrase.*
**Response: This has been removed in the revised manuscript.**

*Page 18 line 362: Update to "fire results in the replacement"*
**Response: This has been changed in the revised manuscript. Please check line 899 in marked-up manuscript version.**

*Page 18 line 363: Update to "grasses in the entire central Cerrrado region"*
**Response: This has been changed in the revised manuscript. Please check line 900 in marked-up manuscript version.**

*Page 19 line 364: Update to: "the importance of considering the interannual climate"*
**Response: This has been changed in the revised manuscript. Please check line 901 in marked-up manuscript version.**

*Page 19 line 369-370: Update to "indicating difficulty in simulating transitional vegetation in these regions"*
**Response: This has been changed in the revised manuscript. Please check lines 969-970 in marked-up manuscript version.**

*Page 22 line 449: Should "according to fire probability" be updated to fire occurence or size? How does the fraction of reduction relate to fire? Is there a fuels component?*

**Response: This has been changed in the revised manuscript. Please check lines 410-432 and 1153-1157 in marked-up manuscript version.**

*Page 23 line 447-453: Suggest moving this section (Sentence beginning "In INLAND, fire…") to Section 2.2, and include one sentence to remind reader of these details. (As stated in section 2.2, fire acts on upper and lower LAI according to fire intensity, triggering competition.)*

**Response: This has been changed in the revised manuscript. Please check lines 1153-1157 in marked-up manuscript version.**

*Page 23 line 458-462: In 458 you state that "the model does not include …fire intensity…" but then state that trees and grasses "are equally exposed to the same fire intensity" Update this so that these statements are consistent. Update line 462 to state what aspect of fire is modifying PFTs.*

**Response: This has been changed in the revised manuscript. Please check line 1154 in marked-up manuscript version.**

*Page 24 line 475-480: "…other important factors are still needed…" improved representation of fire and vegetation fire resistance would also improve these results.*

**Response: This has been changed in the revised manuscript. Please check lines 1272-1278 in marked-up manuscript version.**

*Page 24 line 482: Update "for modeling are peculiar" to "for modeling the transition area are unique."*

**Response: This has been removed in revised manuscript version.**

*Page 25 line 492: Update to "and P have a smaller influence"*

**Response: This has been changed in the revised manuscript. Please check lines 1283-1284 in marked-up manuscript version.**

*Page 25 line 496: Replace "phytophysiognomies distribution" to "vegetation structure and distribution"*

**Response: This has been changed in the revised manuscript. Please check line 1288 in marked-up manuscript version.**

*Page 26 line 514: Update to "indicating that INLAND has difficulty representing the ABG…"*

**Response: This has been changed in the revised manuscript. Please check line 1401 in marked-up manuscript version.**

*Page 26 line 516: Suggest updating "physiognomies" to "vegetation characteristics" or "vegetation types"*

**Response:  This has been changed in the revised manuscript. Please check line 1403 in marked-up manuscript version.**

*Page 26 line 522: Update to "Amazon-Cerrado border are influenced not only"*
**Response:  This has been changed in the revised manuscript. Please check line 1437 in marked-up manuscript version.**

*Page 28 line 553: Update to "lack of measured field parameters"*
**Response:  This has been changed in the revised manuscript. Please check line 1525 in marked-up manuscript version.**

[revised manuscript text omitted]

Formatado: Fonte: Itálico
Formatado: Fonte: Itálico
Formatado: Fonte: Itálico
Formatado: Fonte: Itálico
Formatado: Não Sobrescrito/ Subscrito

**Table 6.** Summary of average NPP, LAI and AGB for the Amazon–Cerrado transition zone over the transect domains, considering all factor combinations. One-way ANOVA results are also shown, including $F$ statistics, and $p$ values. Values within each column followed by a different letter differ significantly ($p < 0.05$) according to the Tukey–Kramer test ($n = 310$: 31 pixels $\times$ 10 years).

| | **NPP** | | **LAI$_{total}$** | | **LAI$_{lower}$** | | **LAI$_{upper}$** | | **AGB** | |
|---|---|---|---|---|---|---|---|---|---|---|
| | kg C m$^{-2}$ yr$^{-1}$ | | m$^2$ m$^{-2}$ | | m$^2$ m$^{-2}$ | | m$^2$ m$^{-2}$ | | kg C m$^{-2}$ | |
| CV+PC | 0.69 | bcd | 6.96 | d | 0.84 | e | 6.48 | a | 9.01 | ab |
| CV+PG | 0.61 | f | 6.24 | f | 0.85 | e | 5.60 | bc | 7.91 | c |
| CV+PR | 0.62 | f | 6.33 | f | 0.85 | e | 5.74 | bc | 8.04 | c |
| CV+PC+F | 0.69 | abc | 7.92 | b | 2.91 | cd | 4.61 | ef | 4.89 | de |
| CV+PG+F | 0.63 | ef | 7.76 | b | 3.73 | a | 5.81 | bc | 3.91 | f |
| CV+PR+F | 0.63 | ef | 7.65 | bc | 3.47 | ab | 4.69 | ef | 4.02 | f |
| CA+PC | 0.72 | ab | 7.39 | c | 0.91 | e | 6.12 | ab | 9.31 | a |
| CA+PG | 0.64 | def | 6.64 | e | 0.91 | e | 5.40 | cd | 8.22 | c |
| CA+PR | 0.65 | cdef | 6.72 | de | 0.91 | e | 5.49 | cd | 8.31 | bc |
| CA+PC+F | 0.74 | a | 8.29 | a | 2.69 | d | 5.02 | de | 5.40 | d |
| CA+PG+F | 0.67 | cde | 7.90 | b | 3.29 | abc | 4.04 | g | 4.45 | ef |
| CA+PR+F | 0.67 | cde | 7.88 | b | 3.19 | bc | 4.18 | fg | 4.42 | ef |
| $F$ | 16.2 | | 115 | | 140 | | 38.1 | | 172 | |
| $p$ | <0.001 | | <0.001 | | <0.01 | | <0.01 | | <0.001 | |

**Table 7**. Correlation coefficients between AGB simulated by INLAND and field estimates ($n$ = 310: 31

pixels × 10 years).

| | T1 | T2 | T3 | T4 | T5 | All transects |
|---|---|---|---|---|---|---|
| CA+PC | 0.843 | 0.928 | 0.886 | 0.937 | 0.337 | 0.786 |
| CV+PC | 0.838 | 0.884 | 0.890 | 0.939 | 0.355 | 0.781 |
| CA+PR | 0.793 | 0.848 | 0.830 | 0.911 | 0.399 | 0.756 |
| CV+PR | 0.795 | 0.793 | 0.832 | 0.907 | 0.527 | 0.771 |
| CA+PG | 0.814 | 0.951 | 0.838 | 0.889 | 0.388 | 0.776 |
| CV+PG | 0.825 | 0.922 | 0.840 | 0.879 | 0.496 | 0.792 |
| CA+PC+F | 0.988 | 0.987 | 0.977 | 0.892 | 0.133 | 0.795 |
| CV+PC+F | 0.976 | 0.947 | 0.933 | 0.908 | 0.187 | 0.790 |
| CA+PR+F | 0.842 | 0.805 | 0.981 | 0.808 | 0.561 | 0.799 |
| CV+PR+F | 0.925 | 0.804 | 0.927 | 0.808 | 0.319 | 0.757 |
| CA+PG+F | 0.844 | 0.961 | 0.980 | 0.830 | 0.430 | 0.809 |
| CV+PG+F | 0.845 | 0.932 | 0.931 | 0.881 | 0.177 | 0.753 |
| CA avg | 0.854 | 0.913 | 0.915 | 0.878 | 0.375 | 0.787 |
| CV avg | 0.867 | 0.880 | 0.892 | 0.887 | 0.344 | 0.774 |